psychology

cognitive development, implicit Theory of Mind, false belief, Sefo, replication, pragmatics

**Authors for correspondence:**
Lisa Wenzel
e-mail: lisa.wenzel@uni-goettingen.de
Sebastian Dörrenberg
e-mail: sebastian.doerrenberg@uni-bremen.de

# Actions do not speak louder than words in an interactive false belief task

Lisa Wenzel[1,†], Sebastian Dörrenberg[2,3,†], Marina Proft[1], Ulf Liszkowski[3,‡] and Hannes Rakoczy[1,‡]

[1]Developmental Psychology, University of Göttingen, Waldweg 26, 37073 Göttingen, Germany
[2]Developmental and Educational Psychology, University of Bremen, Hochschulring 18, 28359 Bremen, Germany
[3]Developmental Psychology, University of Hamburg, Hamburg, Germany

LW, 0000-0003-1761-0588; SD, 0000-0003-3489-5001

Traditionally, it had been assumed that meta-representational Theory of Mind (ToM) emerges around the age of 4 when children come to master standard false belief (FB) tasks. More recent research with various implicit measures, though, has documented much earlier competence and thus challenged the traditional picture. In interactive FB tasks, for instance, infants have been shown to track an interlocutor's false or true belief when interpreting her ambiguous communicative acts (Southgate *et al.* 2010 *Dev. Sci.* **13**, 907–912. (doi:10.1111/j.1467–7687.2009.00946.x)). However, several replication attempts so far have produced mixed findings (e.g. Dörrenberg *et al.* 2018 *Cogn. Dev.* **46**, 12–30. (doi:10.1016/j.cogdev.2018.01.001); Grosse Wiesmann *et al.* 2017 *Dev. Sci.* **20**, e12445. (doi:10.1111/desc.12445); Király *et al.* 2018 *Proc. Natl Acad. Sci. USA* **115**, 11 477–11 482. (doi:10.1073/pnas.1803505115)). Therefore, we conducted a systematic replication study, across two laboratories, of an influential interactive FB task (the so-called 'Sefo' tasks by Southgate *et al.* 2010 *Dev. Sci.* **13**, 907–912. (doi:10.1111/j.1467-7687.2009.00946.x)). First, we implemented close direct replications with the original age group (17-month-olds) and compared their performance to those of 3-year-olds. Second, we designed conceptual replications with modifications and improvements regarding pragmatic ambiguities for 2-year-olds. Third, we validated the task with explicit verbal test versions in older children and adults. Results revealed the following: the original results could not be replicated, and there was no evidence for FB understanding measured by the Sefo task in any age group except for adults. Comparisons to explicit FB tasks suggest that the Sefo task may not be a sensitive measure of FB understanding in children and even underestimate their ToM abilities. The findings add to the

†Shared first authorship.
‡Shared senior authorship.

growing replication crisis in implicit ToM research and highlight the challenge of developing sensitive, reliable and valid measures of early implicit social cognition.

## 1. Introduction

Theory of Mind (ToM) describes our capacity to ascribe mental states like beliefs, intentions, and desires to others and ourselves. At the conceptual heart of ToM lies meta-representation: representing how subjects represent the world, accurately or inaccurately. Traditionally, meta-representation has been measured by explicit verbal tasks that require participants to understand how an agent in a story sees the world and how this differs from the participant's perspective on the world [1]. In the most well-known version of such false belief (FB) tasks, an object is placed in one of two boxes in the presence of the story protagonist, then transferred to the other box in her absence, and children are asked to predict where the agent will look for the object upon her return. While younger children predict that the agent will look for her object in the new location, children from around 4 years pass the task by predicting that she will look for it in the initial box. Decades of research have shown that these results are highly robust and replicable, and that competence emerges in synchrony and correlated fashion across all kinds of superficially dissimilar tasks that all tap meta-representation—suggesting that the 4-year-transition marks a deep conceptual revolution [2,3].

Recently, this traditional view on ToM development has been challenged, and FB research has been revolutionized by studies with new non-verbal, implicit measures with infants (see [4] for a review) and adults (e.g. [5–7]). Violation-of-expectation looking-time paradigms, for instance, showed that infants expected an agent to look for her object where she falsely believed it to be (the infants looked longer in surprise when the agent reached for the box that really contained the object; e.g. [8]). In anticipatory looking paradigms, infants and adults correctly anticipated where an agent, who is mistaken about an object's location, will look for it [5,9] and in priming tasks, adults automatically considered an agent's visual perspective, even though they were instructed only to keep track of their own [10]. In interactive tasks, infants interpreted an agent's behaviour according to her belief in order to help her to achieve a goal [11–13]. These findings on infants' FB understanding and adults' automatic tracking of others' perspectives had far-reaching implications for theoretical approaches to ToM development including nativism [14] and two-systems-accounts [15]. In contrast to conceptual-change accounts, both suggest ontogenetically early emerging and automatic forms of FB ascription abilities. According to nativism, infants operate with a conceptually fully fledged ToM (as indicated in the new implicit tasks) but fail explicit FB tasks because of extraneous linguistic, inhibitory and other performance factors [14]. According to two-systems-accounts, infants do not yet operate with a fully fledged ToM system that develops later and is tapped by explicit tasks, but they do operate with a more basic, early emerging, fast and automatic system for tracking mental states that allows them to pass some implicit tasks [15,16].

Considering these far-reaching implications of the findings from implicit FB tasks and given the recent replication crisis in many areas of psychological science, questions regarding the reliability and replicability of the original results need to be taken seriously. In fact, a growing body of replication studies puts into question the robustness of the original evidence on early FB understanding (for an overview, see [17]; but see also [14] for a review of current successful studies and conceptual replications). The largest body of systematic replication evidence comes from anticipatory looking measures (e.g. [18–21]). Many direct and conceptual replication studies failed to reproduce the original effects with infants, children and adults. Regarding violation-of-expectation tasks with infants, several conceptual replication studies have produced very mixed results, with several failed replication attempts [18,22,23]. Note, however, that there is currently a debate on how serious these replication failures are [24,25].

What about the replicability of interactive measures with infants and children? This will be the focus of the present research. One of the first interactive FB tasks showed that 18-month-old infants adjusted their helping behaviour according to an agent's true belief (TB) or FB [11]. In this task, an agent put her object in one of two boxes. In either her presence (TB) or absence (FB), the object was transferred to the other box. In the test phase of both conditions, the agent approached the empty box, tried (unsuccessfully) to open it, and infants were encouraged to help. In the TB condition, infants approached the empty box more often than the full box, while infants helped to open the box containing the object more often in the FB condition. The original authors' interpretation was the

following: infants understood that the agent was aware of the transfer in the TB condition and thus must have been trying to open the box for some other reason; in contrast, in the FB condition, they understood that the agent was mistaken about the location of the object and was searching for it when trying to open the box. Several follow-up studies have shown the following: first, regarding reliability, the original results seem to be at least partially replicable [22,26–28]. Second, with regard to validity, the original task may not measure belief-tracking but other forms of social cognition such as goal-understanding. According to an alternative interpretation and new empirical data, the task could be mastered by teleological reasoning without any notion of subjective beliefs ([29]; but see [24]). Priewasser *et al.* [29] added another control condition to the Buttelmann task, in which the agent in the FB condition unsuccessfully tried to open a third box that was always empty (neutralizing the agent's false belief). As in the original FB condition, in this new condition, infants still helped the agent by opening the box containing the object.

Another influential interactive FB task that is less ambiguous in its interpretation is the so-called 'Sefo task' by Southgate *et al.* [12]. The basic procedure of the Sefo task is based on referential FB tasks by Carpenter *et al.* [30] and Happé & Loth [31], in which an agent held a FB about the type of object inside a container and requested or named the object she falsely believed to be inside by using a novel label. These studies found that, compared to their performance in standard FB tasks, even 3-year-olds (and older children) performed proficiently in tracking an agent's FB to assign reference in situations that involve learning of new object labels. However, a recent study by Papafragou *et al.* [32] suggests that this effect may be an artefact of miss-matched control conditions: once both task types were closely matched, no performance-enhancing effects of word learning contexts over standard FB tasks were found.

The procedure of the Sefo task was the following: an experimenter (E1) showed two novel objects to the infant, put each object in a separate box, and left the room. Another person (E2) then swapped the objects, which E1 did (TB condition) or did not witness (FB condition). Afterwards, E1 pointed at one box and requested the child to retrieve the object. Southgate *et al.* [12] conducted three experiments that differed in the phrasing of E1's request: while in exp. 1 she asked, 'Do you remember what I put in here? There's a Sefo in here. […] Can you get the Sefo for me?', in exp. 2 the same phrasing without the new object label was used ([…] Shall we play with it? […] Can you get it for me?). Because the phrase 'Do you remember what I put in here?' provides a semantic shortcut to the non-referred box without belief ascription (E1 refers to the object she initially put in that box, but that is now in the other box), in exp. 3 E1 instead asked, 'Do you know what's in here?' plus using the novel label. In all three experiments, 17-month-old infants adjusted their choice of object according to E1's belief, mostly choosing the object in the non-referred box in the FB, and the object in the referred box in the TB condition. These findings suggest that the use of a novel word label and the reference to previous events (semantic shortcut) in the request were not decisive for infants' success in belief-tracking.

Subsequent direct and conceptual replication studies on the Sefo task produced mixed findings (see table 1 for an overview). Of the three published replication studies (for unpublished studies, see [17]), two failed to reproduce the original results [18,34] and one provided a successful extension of the original findings [33]. In a conceptual replication study, Király *et al.* [33] investigated when in an FB story children track an agent's belief and tested 18- and 36-month-olds in scenarios with either prospective or retrospective belief revision phases: E1 placed two novel objects each in one box. While she wore sunglasses, the objects were swapped. Either before (prospective belief revision), or after the swap (retrospective belief revision), children learned that E1's sunglasses were opaque in the FB condition, or transparent in the TB condition. Three-year-old children showed a difference in performance between TB and FB condition in both types of belief revision phases, mostly choosing the object in the non-referred box in the FB, but the object in the referred box in the TB conditions. The 18-month-olds performance was less clear: in the retrospective conditions, they preferred to retrieve the object out of the referred box, with above-chance level performance in the TB and FB condition. In the prospective FB condition, though, two-thirds of infants correctly chose the non-referred box, which differed significantly from the retrospective TB condition. To test their target question, Király *et al.* [33] had to add further demands and performance factors to the Sefo task. For example, keeping track of the sunglasses may raise the working memory load, but may on the other hand also highlight the salience of the actor's perspective. These modifications, and the fact that 18-month-olds received two attempts to pass the task of which they failed one, make it difficult to interpret their findings (the positive and the negative ones) in light of replicability of the original study because we do not know which factors might have boosted or diminished performance.

Dörrenberg *et al.* [18] tested 2-year-old children with the original procedure and failed to replicate the results of a significant difference between the conditions. Here, children chose the referred box more often

**Table 1.** Studies using the Sefo task with age group, sample size and results.

| study | age group and sample size | results |
|---|---|---|
| Southgate et al. [12] | 17-month-olds (n = 60) | significant difference in the choice of boxes between FB and TB: predominant choice of non-referred box in the FB condition, predominant choice of referred box in TB condition |
| Király et al. [33] | 18-month-olds (n = 46) | no differences in choice of boxes between FB and TB condition in retrospective task: in both conditions, infants chose the referred box predominantly performance in prospective FB task differed to retrospective TB performance |
| | 3-year-olds (n = 72) | differences between FB and TB condition in both task types pro- and retrospective: predominant choice of non-referred box in the FB condition, predominant choice of referred box in TB condition |
| Dörrenberg et al. [18] | 24-month-olds (n = 60) | no differences in choice of boxes between conditions: chance level performance in FB condition, significant choice of referred box in TB condition |
| Grosse Wiesmann et al. [34] | 3-year-olds (n = 26) and 4-year-olds (n = 31) | both age groups performed at chance level in FB condition no TB condition tested |

than expected by chance in the TB condition and at chance levels in the FB condition. Grosse Wiesmann et al. [34] tested 3- and 4-year-old children with the FB condition and also failed to replicate the original above-chance performance (note that a corresponding TB condition was missing). However, both replication studies had different methodological limitations that could have led to the low performance. In particular, the children tested in both replication studies were older than in the original study. It is possible that this kind of task is not equally suitable for older children. Thus, in order to stringently investigate the task's reliability, replication studies with the original age group are needed. The study by Grosse Wiesmann et al. [34] added further modifications. First, E2 was present during the whole scenario, not hidden as in the original study. Grosse Wiesmann et al. [34] discussed in their paper as a limitation of their study, that the continuous presence of E2 may have diminished E2's deceptive motive. That is, because deceptive behaviour can enhance young children's performance in explicit FB tasks [3,35], it is possible that the missing deceptive motive could have led to the low performance in the FB condition. Second, explicit test questions were introduced. After the child chose one of the objects, E1 explicitly asked which one of the two objects the Sefo was and with which she had wanted to play with. Here, 3-year-olds performed at chance level, while 4-year-olds showed above-chance performance. Weak correlations were found between children's performance in the original Sefo task and their performance in the explicit test questions and an additional verbal standard FB task, suggesting a unified explicit ToM competence underlying the Sefo task.

A concern regarding recent replication studies of the Sefo task is that older children's growing linguistic abilities have not been taken into account sufficiently. As already mentioned, the phrasing of the request from Southgate et al. [12] exps 1 and 2 (Do you remember what I put here?), provides a semantic shortcut to passing the FB condition without belief ascription. That is, since the objects were swapped, the correct answer to this question would always be to choose the object out of the other, non-referred box. While this semantic explanation leads to the pattern found in the FB condition, it would not predict the pattern found in the TB condition (here choosing the referred box is correct); however, in the TB condition, other plausible strategies may come into play (e.g. obeying the pointing gesture without paying attention to E1's request). Because E1 witnessed everything that has happened, the situation is very clear for her and the child. Although this request may not be an issue when testing infants (see Southgate et al.'s exp. 3), older children's linguistic abilities make it likely

that they understand the experimenter's verbal prompt and solve the FB condition without any belief ascription. Accordingly, semantic hints in the experimenter's request may have boosted FB performance in the study by Király *et al.* [33] that used the original phrasing from exp. 1 (Do you remember what I put here?) for testing 3-year-olds. The study by Grosse Wiesmann *et al.* [34] used a new phrasing: 'Do you still know what is in this box? There is a Sefo in here'. In contrast to the original requests, by using a literal interpretation of this new request, older children might be biased to the object in the referred box (incorrect in FB condition), which would explain the poor FB performance. Accordingly, especially for testing older children with growing linguistic abilities, using a neutral phrasing of the request appears particularly important.

In summary, to this date, three published replications of the Sefo task have produced a very mixed set of findings. Different age groups and/or methodological alterations may have influenced and masked children's performance. In order to investigate the reliability and robustness of the original findings, more systematic and comprehensive replication approaches are thus necessary. Therefore, the aim of the present study was to implement a collaborative replication project of the 'Sefo task' across two laboratories. In order to investigate the replicability, robustness, and validity of the original findings, direct and conceptual replications were conducted in different age groups from 17-month-olds to adulthood. In study 1, a direct replication study, we replicated the original Sefo task as closely as possible with the original methods with both the original age group and 3-year-olds. Furthermore, 3-year-olds' performance was compared to their performance in a standard FB task, to investigate whether both tasks tap the same competence. Because the original Sefo task suffers from equivocality and pragmatic ambiguities, we designed two modified and improved task versions in conceptual replication studies with 2-year-olds in study 2. Finally, study 3 was a validation study in which we investigated whether the Sefo tasks measures what it is supposed to measure, i.e. belief-tracking. To this end, we tested for the convergent validity of the Sefo task with established FB tasks in older children and adults.

# 2. Study 1: direct replication

The aim of the first study was to directly replicate the Sefo task. Therefore, we used the same method and the same procedure as in the original study to test the original age group (17-month-olds). Additionally, we explored potential learning effects by administering multiple test trials and offering feedback after every trial. To investigate developmental trajectories and potential relations to explicit ToM abilities, we tested an additional older age group (3-year-olds) with the original Sefo task and a standard FB (change-of-location) task.

## 2.1. Methods

### 2.1.1. Participants

The final sample consisted of forty-eight 17-month-olds (twice the sample size of each experiment of the original study; median age = 17 months; 14 days, age range = 16;11–18;2, 25 girls and 23 boys) and forty-eight 3-year-olds (median age = 41.5 months, age range = 36–47 months, 27 girls and 21 boys). Half of the sample was tested in Göttingen, the other half in Hamburg (the TB condition of the 3-year-olds was tested only in Göttingen). Participants were recruited from databanks of children whose parents had previously agreed to participate in child studies. Twenty-six more children were tested but excluded because of refusal to participate (17-month olds: $n = 14$/3-year-olds: $n = 1$), failing the warm-up trials (17-month-olds: $n = 9$), parent interference (17-month-olds: $n = 1$) or experimenter error (3-year-olds: $n = 1$).

### 2.1.2. Design and procedure

In both age groups, we tested FB and TB conditions in a between-subjects design. Seventeen-month-old participants received three Sefo test trials and corresponding feedback after every trial. Three-year-olds were tested with a single Sefo test trial (either TB or FB) and one trial of a standard (change-of-location) task in the same condition (task order counterbalanced).

#### 2.1.2.1. Sefo task

Children were seated on the floor between their parent's legs. Two boxes (lids attached so they remained in an upward position when opened) were positioned 120 cm from the infant and 100 cm apart facing the

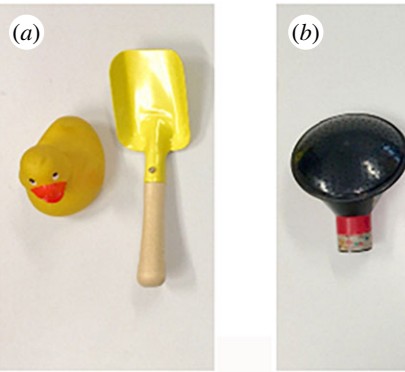
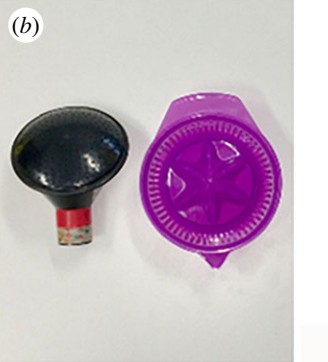
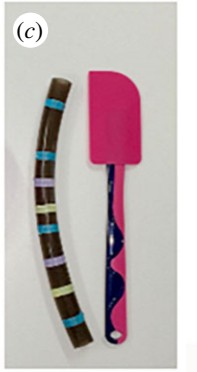
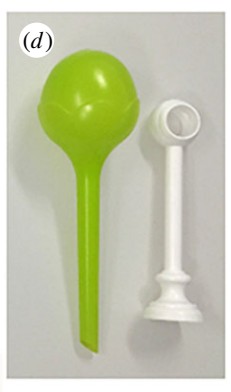

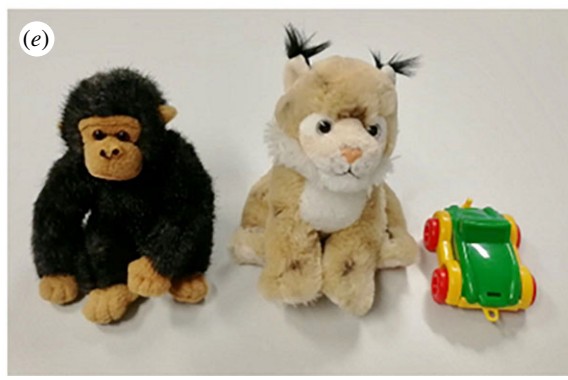

**Figure 1.** Objects used in the direct Sefo task replication (*a–d*; (*a*) for familiarization trials, (*b*) for first trial of 17-month-olds, (*c*) and (*d*) for second and third trial of 17-month-olds (counterbalanced), (*d*) for 3-year-olds) and in the SBT (*e*) in study 1.

child. Experimenter 2 (E2) hid behind curtains. Before the test trials, at least two warm-up trials were conducted: participants were allowed to explore two familiar objects (a bathing duck and a toy shovel) for about 10 s. Afterwards, experimenter 1 (E1) put one object in each box and closed the lids. Then E1 asked the child to bring her one of the objects. This was repeated until the participant brought each object once in two consecutive trials (original inclusion criterion). Note, that some 17-month-olds had difficulty mastering the warm-up phase. Thus, to reduce the number of dropouts, we applied a more lenient passing criterion (bringing both objects needed not to be in consecutive trials). In addition, we had to modify the warm-up procedure slightly for some participants (e.g. leaving the boxes open when requesting an object).

In the test trials, E1 presented two novel objects (a water can spout and a lemon squeezer in the first trial for the 17-month-olds as in the original study, and new object pairs for the other trials; a curtain holder and a plant watering bulb for the 3-year-olds (objects adjusted to older age); figure 1), allowed the child to explore them for about 10 s, placed each in a box and closed the lids. Then E1 left the room and E2, hitherto unknown to the child and hidden behind curtains, entered the scenery in a deceptive manner (alternating gaze between the door and the child). E2 sat down between the boxes and swapped the objects: E2 opened both boxes, placed one object in front of its box, took the other object, showed it to the infant, placed in the other box, picked up the first object, showed it to the infant, placed it in the other box, and closed both boxes simultaneously. Throughout her presence, E2 emphasized her deceptive plan, e.g. by whispering, giggling, and gesturing, 'Shush.' In the TB condition, E2 also appeared in a deceptive manner, but stopped acting deceptively when E1 reappeared. This happened before E2 started to swap the objects and E1 sat down next to the child. During the swap, E1 commented on E2's actions ('Ah, okay!') to emphasize that she witnessed all relevant events. In the FB condition, however, E1 re-entered the room shortly after E2 hid behind the curtains again. From here on, TB and FB conditions again follow the very same procedure: E1 sat down between the boxes in a position from where she could not look inside and asked the infant, 'Do you know what's in here? I want to play with this!' while tapping at one of the two boxes (side and target object counterbalanced). E1 opened both boxes simultaneously, faced the infant, and asked, 'Can you give it to me?' E1 repeatedly requested the object until participants pointed at or approached a box. In order to avoid possible alternative interpretations and to keep the request as

**Table 2.** Number of children that chose the referred/non-referred box in the first test trial with the applied warm-up criterion (original or modified) as a function of the warm-up version (original or modified).

| | | original criterion | modified criterion |
|---|---|---|---|
| TB | original warm-up | 2/1 | 1/0 |
| | modified warm-up | 8/5 | 5/2 |
| FB | original warm-up | 2/2 | 1/0 |
| | modified warm-up | 5/3 | 7/4 |

simple as possible, we used the first part of the phrasing from Southgate *et al.*'s exp. 3 that did not refer to the child's memory of previous object locations (thus did not provide a semantic shortcut). Note that we did not include the novel label for the referred object in the request, because Southgate *et al.*'s study showed that it is not necessary for infants' success (see also [32]). We coded for the box that participants approached or pointed at first, which was either the referred box (correct in TB, incorrect in FB) or the non-referred box (incorrect in TB, correct in FB). In addition, we coded whether children informed E1 by commenting on E2's presence (e.g. by pointing at E2's hiding place). For further alternative dependent measures that are not part of the replication analysis see Appendix A.

After offering E1 an object, the 17-month-olds received helpful feedback. When they chose correctly (TB: referred box, FB: non-referred box), E1 acted happy about the chosen toy (in FB also surprised about the new location). When they chose incorrectly (TB: non-referred box, FB: referred box), E1 acted surprised and said, 'Humph. Strange. No, that is not what I meant.' She then looked in the other box and said, 'Ah. It is in here. Okay.' In each case, they briefly played with the correct object.

### 2.1.2.2. Standard belief task

In the standard belief task (SBT), a standard change-of-location task, the participants were positioned as in the Sefo task (but two containers were 50 cm apart and 50 cm away from the child) and E1 acted the story of a cuddly toy lynx (Luchsi) on the floor. Luchsi showed her toy car to the child and played with it briefly. Then she put her car into one of two little boxes and left the scene. In the absence of Luchsi (FB), or before she left (TB), an ape puppet appeared, swapped the car to the other box and left. E1 then asked three control questions (Where did Luchsi put his car in the beginning?, Where is the car now? and Who put it there?). Children got corrective feedback when answering the control questions incorrectly. Then, E1 asked an explicit test question (When Luchsi returns, where will Luchsi look for her car first?). Children could either indicate that Luchsi will search in the box containing the object (correct in TB, incorrect in FB) or in the empty box (incorrect in TB, correct in FB).

## 2.2. Results

### 2.2.1. Direct replication

Table 2 shows the number of children choosing the referred and non-referred box in their first test trial after having received either the original or the modified warm-up as a function of the applied familiarization criterion. Overall, nine infants received the original warm-up version, while 39 were tested with the modified version. To ensure that the modifications of the warm-up phase had no influence on the performance in the interactive Sefo task, we compared the first test trial performance of those children who received the modified version with those who received the original version. We did not find differences in FB or TB conditions (Fisher's exact tests, both $ps = 1$). Furthermore, we compared the choice of box between those participants who only met the more lenient criterion (FB: $n = 12$, TB: $n = 8$) and those who passed the original criterion (FB: $n = 12$, TB: $n = 16$). There was no difference in the choice of box between the groups in each condition (Fisher's exact tests, FB condition: $p = 1$, TB condition: $p = 0.667$), and within each group no difference between FB or TB condition (Fisher's exact tests, original criterion: $p = 1$, modified criterion: $p = 1$). Thus, we collapsed all participants for further analyses. To provide a direct replication of the analyses of Southgate *et al.* [12] who used a single-trial design, we first present analyses of the first test trial for the 17-month-olds.

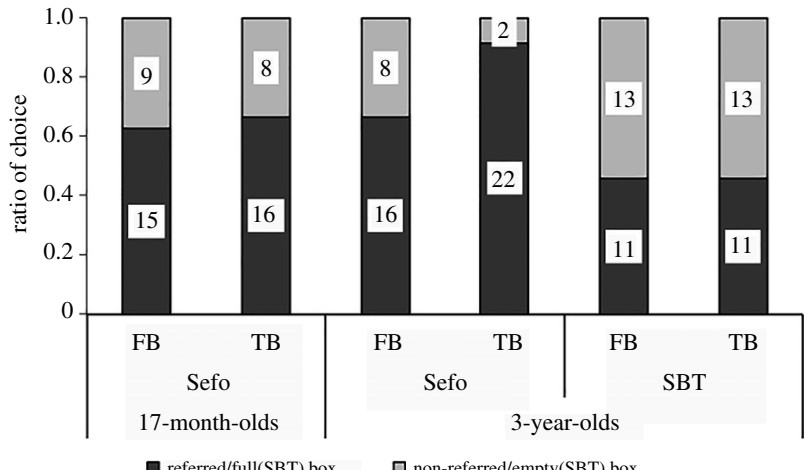

**Figure 2.** Proportion of participants (17-month-olds (first trial performance depicted) and 3-year-olds) who chose each box in the false belief (FB) and the true belief (TB) condition in the direct replication of the Sefo task and SBT in study 1. Numbers in bars show number of participants.

**Table 3.** Correct performance (%) of those infants ($n = 28$) who received all three trials of the Sefo task in study 1 as a function of condition and trial.

|     | trial 1 | trial 2 | trial 3 |
| --- | --- | --- | --- |
| FB  | 27 | 60 | 27 |
| TB  | 54 | 69 | 54 |

### 2.2.1.1. First trial

Seventeen-month-olds' performance in the first trial is depicted in figure 2. In the FB condition, 15 of 24 (62%) incorrectly chose the referred box (binomial tests: $p = 0.307$); in the TB condition, 16 of 24 participants (67%) correctly chose the referred box (binomial test, $p = 0.152$). In contrast to the original study, there was no difference in proportion of infants who chose the referred box between the TB and the FB condition (Fisher's exact test, $p = 1$). There were no effects of sex or laboratory (Fisher's exact tests, all $ps \geq 0.667$).

Of the 24 three-year-olds, in the FB condition of the Sefo task, 16 children (67%) incorrectly chose the referred box, which was not different from chance (binomial test, $p = 0.152$). In the TB condition of the Sefo task, 22 of 24 (92%) correctly chose the referred box, which differed significantly from chance (binomial test, $p < 0.001$). The 3-year-olds tended to choose differently in the two conditions of the Sefo task (Fisher's exact test, $p = 0.072$). A comparison of performance in the Sefo task between 17-month-olds (first trial) and 3-year-olds showed no difference in the FB condition (Fisher's exact test, $p = 1$). In the TB condition, 3-year-olds tended to perform better compared to 17-month-olds (Fisher's exact test, $p = 0.072$).

In the 17-month-olds, we found a marginal difference between the conditions regarding informing behaviour (Fisher's exact test, $p = 0.701$). Four of the twenty-four 17-month-olds (17%) informed E1 after her return in the FB condition about E2's presence, by e.g. pointing towards the curtains or indicating that the objects have been swapped. In the TB condition, three of twenty-four 17-month-olds informed E1 after E2 disappeared behind the curtains, even though she knew about her presence. Three-year-old children informed E1 significantly more often in the FB condition than in the TB condition (Fisher's exact test, $p = 0.022$). While six of 24 children (25%) showed informing behaviour in the FB condition, none of them did so in the TB condition. Of the six children who informed E1, five chose the non-referred box, which was significantly more often compared to those who did not inform E1 (Fisher's exact test, $p = 0.007$).

### 2.2.1.2. Repeated trials in 17-month-old children

In total, 58% of children performed all three test trials (FB: 15 participants, TB: 13 participants; 20 children failed at least one trial owing to refusal of participation). Table 3 shows the performance of these children as a function of test trials. Within each condition (FB or TB), there was no difference in performance between the three test trials (Cochran's Q-tests, FB: $\chi^2_2 = 4.17$, $p = 0.125$; TB: $\chi^2_2 = 0.8$, $p = 0.67$). Within

**Table 4.** Contingency table of 3-year-old's performance in both tasks (Sefo and SBT) in either TB or FB condition.

| | SBT | |
|---|---|---|
| | empty box | box containing the object |
| Sefo | | |
| TB | | |
| non-referred box | 1 | 1 |
| referred box | 12 | 10 |
| FB | | |
| non-referred box | 3 | 5 |
| referred box | 9 | 7 |

each test trial, there was no significant difference in performance between FB and TB condition (Fisher's exact tests, all $ps \geq 0.15$).

### 2.2.2. Comparison of 3-year-olds' performance in the Sefo task and the standard belief task

Table 4 shows children's consistency over the different tasks in either TB or FB condition. In the FB condition of the standard task, 11 of the 24 children (46%) incorrectly chose the full box, which was not different from chance (binomial test, $p = 0.839$; figure 2). In the TB condition of the standard task, 11 of 24 (46%) correctly chose the full box, which was not different from chance (binomial test, $p = 0.839$). The standard task conditions were not different from each other regarding children's choice of full versus empty box (Fisher's exact test, $p = 1$). A comparison of performances between the Sefo task and the standard task showed no significant difference and no correlation in the FB condition (McNemar test, $p = 0.267$; $\phi_{24} = -0.059$, $p = 0.772$) but a significant difference and no correlation in the TB condition (McNemar test, $p = 0.003$; $\phi_{24} = -0.025$, $p = 0.902$). There were no effects of sex, laboratory or task order (Fisher's exact tests, all $ps \geq 0.729$), and no correlation with age (Sefo: $r_{48} = -0.012$, $p = 0.933$; standard: $r_{48} = -0.024$, $p = 0.873$) for the choice of box in each task.

### 2.3. Discussion

The main findings of study 1 were the following: first, the central original finding that infants respond differently in FB and TB condition could not be replicated. In the present study, infants chose both boxes at chance levels in both FB and TB condition and overall tended to choose the referred box. Second, there was no evidence for learning throughout the multiple trials with feedback. Third, 3-year-old children tended to choose differently in the two conditions of the Sefo task, though they remained at chance in the FB condition. Further, they performed at chance in the explicit SBT, with no difference between conditions. Fourth, there were no correlations between the two tasks in 3-year-olds. Children who passed the SBT did not also pass the Sefo task. Therefore, it remains unclear whether the Sefo task really taps belief ascription abilities like the SBT.

The interpretation of these results is complicated by several factors: regarding the 3-year-olds, the absence of a relationship between Sefo and standard FB task seems difficult to understand given the fact that the 3-year-olds performed at chance in both standard FB and TB tasks. What does this mean? Were children simply guessing in both kinds of tasks? Normally, wouldn't one expect ceiling performance in TB conditions (a kind of baseline control) and below chance or chance performance in this age range? Well, not necessarily. Recent research has found a striking developmental pattern such that children initially pass TB and fail FB task, then begin to show the reverse pattern from around age 4, and only come to master both kinds of tasks from around age 8–10 [36–38]. These findings reveal a U-shaped curve in children's TB development with massive negative correlation to their FB performance. However, Rakoczy & Oktay-Gür [39] showed that these findings are owing to pragmatic confusion instead of competence limitations, and that TB performance can be enhanced with suitable task modifications. It may thus be that the present sample of 3-year-olds contained different subsamples of children: one sub-sample showing the typical 3-year-old pattern (pass TB, fail FB) and the other sub-sample showing the reverse pattern (fail TB, pass FB). At the group level performance would

then look like guessing, but it would be based on systematic answer patterns of two different groups. In line with previous findings [40] this seems a very plausible possibility. The link between 3-year-olds' informing behaviour to their performance in the FB condition revealed that at least some children were able to track E1's state of knowledge and to react appropriately. However, given the between-subjects design of the present study, we cannot tell conclusively.

Another complication comes from younger children's performance in the familiarization phase: 40% of the 17-month-olds tested in study 1 struggled to pass the original familiarization criterion. Is it possible that we did not replicate the original results because of infants' lower performance in the familiarization phase? Maybe infants did not understand the procedure of the Sefo task in general and thus performed at chance level in the test situation. However, we did not find performance differences between the subsamples that did versus did not reach the familiarization criterion. Thus, even those infants who passed the original familiarization criterion did not perform as would have been expected from the original findings. Furthermore, all 3-year-old children passed the original familiarization criterion and still performed at chance level in the FB condition. Therefore, the low performance in the familiarization trials cannot sufficiently explain the low performance in the test scenario and thus cannot account for the failed replication of the original results. Did the difficulties in passing the original familiarization criterion perhaps reflect poor rapport with the experimenter? This appears unlikely given the efforts we made to warm-up with the infants before any testing began. First, infants were not opting out or fussing, which typically happens when they do not feel comfortable with a stranger. Second, all of them interacted with the experimenter during familiarization, when they brought the requested object in two trials (just not two consecutive trials), and in three test trials, which are clear indications of good rapport.

# 3. Study 2: conceptual replication with pragmatic modifications

The aim of the second study was to modify the Sefo task regarding potential pragmatic ambiguities and to investigate whether these modifications improve children's performance. Therefore, we tested 2-year-old children with different versions of the Sefo task, including the original procedure. In the original Sefo task, the procedure may be pragmatically ambiguous and thus difficult to understand in various respects: first, it remains unclear why the experimenter does not get the object by herself. Even though the task is introduced as a fun game in which the infant's task is to retrieve objects out of the boxes, the experimenter was able to reach for the boxes before she had left. This concern was addressed in study 2a by changing the situation in such a way that the experimenter now requested an object she could not reach herself. Second, which object the child is supposed to give in response to the request appears arbitrary and rather irrelevant. Basically, there is no obvious reason why the interlocutor would need one rather than the other object. Third, it appears unclear why E1 leaves the scene and whether her interest in the toys remains stable during her absence. These two concerns were additionally addressed in study 2b in which the set-up was modified in such a way that it was much more relevant which object E1 was looking for and that E1 had good reasons to leave. Here, we again administered multiple trials to test for potential learning effects and varied whether children received corresponding feedback after their choice or not.

## 3.1. Study 2a

In study 2a, we modified the test situation in such a way that E1's request for an object made pragmatically more sense. Rather than asking for an object that she could easily take herself, she now requested one of two objects that, for physical reasons, she could not reach herself. In addition to this modified Sefo task, 2-year-olds also received the original version so that performance could be compared across task versions.

### 3.1.1. Methods

#### 3.1.1.1. Participants

Forty 2-year-old children (median age = 30 months, age range = 24–35 months, 24 girls, 16 boys) from mixed socio-economic backgrounds were tested in the facilities of the University of Göttingen. All participants were recruited from a database of children whose parents had previously given consent

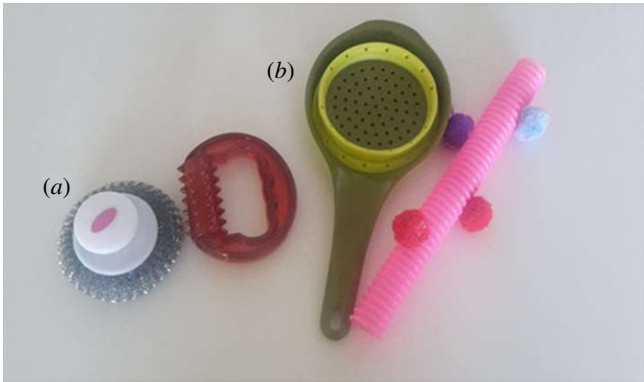

**Figure 3.** Novel objects used in study 2a (($a$): metal scouring pad and massage roller, ($b$): pink tube with glued felt balls and sieve).

to experimental participation. In total, six children were tested but excluded because they refused to participate ($n = 4$), or experimenter error ($n = 2$).

### 3.1.1.2. Material

In each trial, one pair of novel objects was used (counterbalanced order, depicted in figure 3) while the same different-coloured boxes were used in both trials (pink and yellow colour, counterbalanced side). To ensure that children were able to open the boxes by themselves in the modified version of the Sefo task, curtains were installed in front of each box. The study was conducted in a room with two separate doors, one of which was blocked by a table, so that entering the room through this door was impossible.

### 3.1.1.3. Design and procedure

In total, children received one original Sefo task and one modified task with the type of belief (TB/FB), and order of task type (original/modified) as between-subjects factors. The objects' side of placement and the side of pointing were counterbalanced across trials and participants.

Before the test session started, E1 introduced and demonstrated the unbreachable obstacle by trying to enter the test room through the blocked door. She opened the door with the child, expressed her surprise about the blocked way, and explained to the child that they have to use the other door in order to get into the room to the toys because this way seemed impassable. After entering the room successfully through the unblocked second door, the original familiarization phase as described in study 1 followed. To encourage children to pass the requested object over the table, an additional familiarization phase was conducted. E1 placed each object in one box, lowered the curtains, and left the room. After entering the room through the blocked door, she claimed that she took the wrong door by mistake and asked the child for help to pass her one of the objects, the referred one, over the table, and afterwards the other one. All participants were able to bring the referred object in the three familiarization trials.

After the familiarization phases, every child received two test trials either in the FB or TB condition. The direct replication of the Sefo task followed the original procedure as closely as possible (described in study 1), except for the request used by E1 (see detailed description below). The basic procedure of the modified Sefo task is depicted in figure 4. After two novel objects were presented to the child, E1 placed each object in one of two boxes, lowered the curtains of both boxes simultaneously, and left the room through the unblocked door. In the FB condition, the second experimenter (E2) appeared behind a curtain in a secretive manner and sat down between the two boxes facing the child while E1 was absent. In the TB condition, E1 came through the blocked door after E2 appeared and thus witnessed her actions. E2 took the first object out of the box and placed it in front of the box, took the second object out of the box, showed it to the child, placed it in the first box; took the first object, showed it to the child and placed it in the second box. She lowered the curtains of the boxes simultaneously. With a gesture of secretiveness (index finger on her lips and 'Shush'-sound), she left behind the curtain. E1 opened the second, blocked door, claimed that she had forgotten that this way was barred again, and requested the child to help her by passing her the 'Sefo': either after the swap (TB

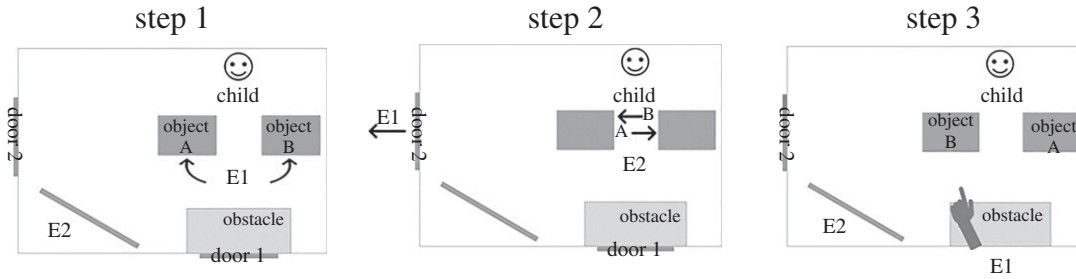

**Figure 4.** Basic set-up of the modified Sefo task in study 2a. Step 1: E1 puts two novel objects, each in one box while E2 is hidden behind a curtain. Step 2: after E1 left through door 2, E2 swaps the two objects. Step 3: E1 enters the room through door 1, points to one of the boxes and requests the infant for help.

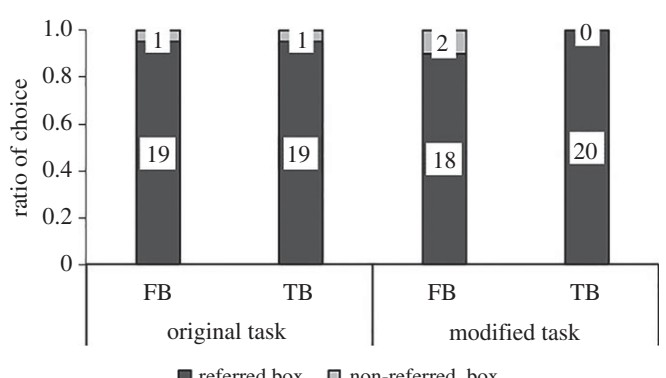

**Figure 5.** Proportion of participants who chose each box in the false belief (FB) and the true belief (TB) condition in the original and the modified Sefo task in study 2a. Numbers in bars show number of participants.

condition) or after E1's return (FB condition), E1 continuously pointed to one of the boxes (side and target object counterbalanced) while saying 'Do you know what is in there? There is a Sefo in this box! A Sefo!' Without pointing she then asked: 'Do we want to play with the Sefo? Can you pass me the Sefo?' (request of Southgate *et al.*'s exp. 3). To keep the two versions as parallel as possible, we stated the very same request also in the original version of the Sefo task. In order to avoid learning effects, we did not give feedback after the first trial. After receiving the 'Sefo', E1 thanked the child and stated that she had two more exciting toys she wanted to show to the child and went on with the demonstration of the second pair of novel toys. Because two consecutive trials were conducted we used the novel label 'Sefo' in the first trial and 'Toma' in the second trial.

### 3.1.2. Results

The proportions of children choosing the different boxes in TB/FB as a function of task (original versus modified) are depicted in figure 5. In both versions of the Sefo task (original and modified), children predominantly chose the box E1 was referring to, irrespective of condition. In the FB condition of the original Sefo task, as well as in the TB condition, 19 out of 20 children (95%) chose the referred box. Thus, performance differed significantly from chance in both conditions (FB and TB: binomial tests, $p < 0.001$). In the FB condition of the modified version of the Sefo task, 18 out of 20 children (90%) chose the referred box, while all 20 children chose the referred box in the TB condition. In both conditions, performance differed significantly from chance (FB and TB: binomial tests, $p < 0.001$). Overall, we did not find any differences in performance between the two versions of the Sefo task (McNemar test, $p = 1$).

## 3.2. Study 2b

study 2a failed to show that 2-year-old children consider an agent's belief in the original Sefo task as well in our modified version. We did not find any differences between conditions, nor between task type. It thus seems that the negative results of our replication attempts are not merely owing to the

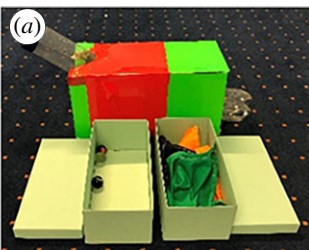
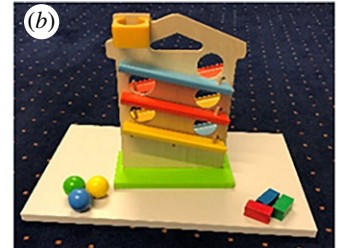
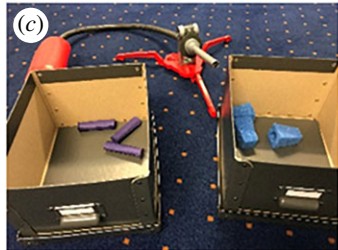
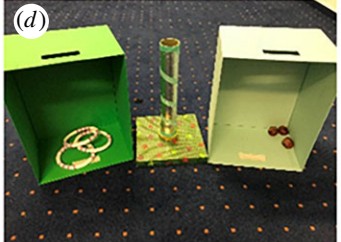
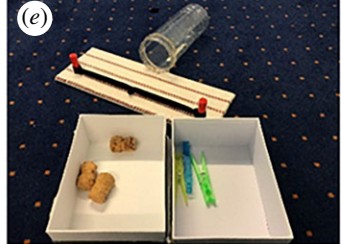

**Figure 6.** Objects and apparatuses used in the familiarization trials (*a,b*) and test trials (*c–e*) of the pragmatically modified Sefo task in study 2b.

pragmatic oddity of the original request (of an object that the experimenter could easily grasp herself). Study 2b addressed two more potential concerns with the original procedure: it increased the relevance of giving the experimenter exactly the right kind of object, and it made E1's leaving the room more plausible. Again, multiple trials were implemented to test for potential learning effects and we counterbalanced between subjects, whether or not children received feedback after trials.

### 3.2.1. Methods

#### 3.2.1.1. Participants
Thirty-six 24-month-olds (median age = 24;20, age range = 23;30–25;12, 17 girls and 19 boys) were tested in the metropolitan city Hamburg, recruited from a databank of children whose parents had previously agreed to participate in infant studies. Two more infants were tested but excluded because they refused to participate.

#### 3.2.1.2. Material
We used a new set of materials for each trial, each consisting of two boxes that each contained different objects (boxes always contained three objects of the same kind), and an apparatus that could be used with the objects from the boxes for a fun game. In the familiarization trials, only the objects from one of the boxes worked with the apparatus. In the test trials, objects from both boxes would potentially work with the apparatus, though it was not obvious which object type would be the match.

Figure 6 depicts all materials. In one familiarization trial, the apparatus was (*a*) an open plastic bottle that contained a chime and was installed diagonally through a box, which could be used with the marbles from one box (running down the bottle, eliciting the chime) but not with the cloths from the other box. In the other familiarization trial, it was (*b*) a ball run where the balls from one box could run down but the bricks from the other box would not work. For the three test trials, we used three different sets: (*c*) bellows that could be used to shoot either purple paper shucks or blue pieces of sponge, (*d*) an upright tube on a board with a rattle on the bottom where chestnuts could be thrown in to elicit the rattle or rattling plastic rings could be thrown over, and (*e*) a slingshot that could shoot either wine corks or clothespins into an attached plastic bottle. Order of material sets and target objects was counterbalanced.

#### 3.2.1.3. Design and procedure
In a between-subjects design, children were randomly assigned to one of three conditions: TB, FB feedback, or FB no feedback. The basic set-up was similar to in the original Sefo task, but we introduced an additional apparatus, which resolved several pragmatic issues: The experimenter had to leave the room to get the apparatus (therefore, making her absence plausible) and it created a fun

**Table 5.** Basic differences between the original Sefo task and the modified task version designed in study 2b.

| original Sefo task | modified Sefo task |
|---|---|
| *familiarization* | |
| two (sets of) objects are placed each in one box | |
| | familiarization of E1 leaving the room to get an apparatus |
| E1 asks the child to find one of the objects | E1 cannot get the object because of the huge apparatus in her hands and asks the child for help to get the referred object |
| | E1 shows importance of infant's object choice: referred object creates a fun game with the apparatus |
| E1 asks the child to find the other object | E1 shows that the non-referred object does not work with the apparatus |
| *test phase* | |
| two (sets of) new objects are placed each in one box | |
| E1 leaves the room without a reason | E1 leaves the room to get another apparatus |
| E2 swaps the objects | |
| E1 comes back and sits down between the boxes, | E1 comes back with the huge apparatus in her hands, asks the child for help, |
| points to one of them and requests the infant to get the object out of the referred box | points to one of the boxes and requests the infant to get the object out of the referred box |

game with the Sefo (therefore, making the infant's choice of the right object relevant). Table 5 shows the main differences between the original Sefo task procedure and the modified task version. Each session started with two familiarization trials, followed by three test trials in the same condition. Already in the familiarization phase, participants learned that E1 leaves the room to get an apparatus and that only one specific set of objects can be used for a game with the apparatus while the other set of objects would not work. This should ensure that in the test trials infants would understand the specificity of E1's request. Boxes remained open throughout a trial after exploring the content (lids lying behind).

First, E1 and the child explored the content of the two boxes. E1 then said, 'I know what we can play with this. I go out and get something for us'. Before E1 left the room, he checked the content of each box again and said, 'Okay, that is in here, and that is in here'. In familiarization trials, E1 returned with an apparatus, slowly walked towards the boxes (centred between the boxes), pointed at one box (acting as if she was struggling to hold the apparatus, stressing the need for help) and said, 'We need the (toy name; e.g. the balls). Can you give it to me?' After an infant gave E1 the correct toys, the two together used them to play with the apparatus (e.g. roll a ball down the ball run for three times). E1 then showed the infant that the other object type would not work with the apparatus (e.g. the bricks would not roll down). All infants correctly brought the indicated objects in the two familiarization trials.

The test trials were similar to the familiarization trials, but E2 entered the room from behind the curtains after E1 left. E2 looked at the infant said, 'Hello' and looked at the door to ensure that E1 was absent. In the TB condition, E1 would now re-enter the room with the apparatus in his hands, greet E2, and position beside the set-up to observe E2. In the FB conditions, E1 was still absent at this point and E2 acted in a sneaky manner (e.g. whispering, giggling, looking at the door). E2 went between the boxes, took out the objects from one box and placed them in front of the box, then she took out the objects from the other box and showed them to the child (and E1 in TB condition), then she swapped the objects from one box to the other and said, 'Look, I put this here'. In the TB condition, E1 commented, 'Ah, okay'. E2 then said goodbye and disappeared behind the curtains again. Subsequently, E1 re-entered the room through the door with the apparatus in his hands in the FB condition, or positioned behind the boxes in the TB condition, respectively. E1 slowly walked towards the boxes (centred between the boxes), pointed at one box (side counterbalanced; acting as if he was struggling to hold the apparatus, stressing the need for help) while saying, 'We need what is in there. Can you give it to me?' He stopped behind the boxes and waited until the infant made a choice.

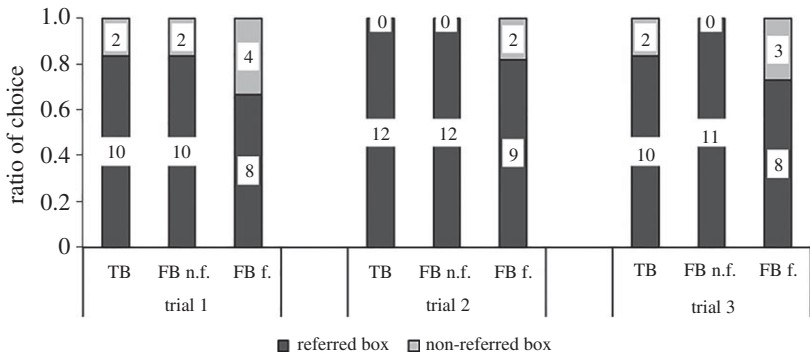

**Figure 7.** Proportion of participants who chose each box in the three test trials for the true belief (TB), false belief no feedback (FB n.f.) and false belief feedback (FB f.) condition in the pragmatically modified Sefo task, study 2b. Numbers in bars show number of participants.

In the FB no feedback condition and in the TB condition, right after the infant gave an object to E1 (correct or incorrect) someone knocked at the door, E1 went to the door and acted as if he would be talking to someone. When he returned to the child, he put away the materials, claimed to have even better toys, and started a new trial. In the FB feedback condition, if the child correctly offered the objects from the non-referred box, E1 acted surprised about the new location, but they played with the toys (e.g. shooting the three sponges with the bellows). If the child incorrectly offered the objects from the referred box, E1 acted surprised and said, 'Humph. Strange. No, that is not what I meant'. He then looked in the other box and said, 'Ah. That is what I meant. How did it get here?' and they played with the objects from the non-referred box.

### 3.2.2. Results

#### 3.2.2.1. First trial
The proportions of children choosing the different boxes as a function of condition and trial are depicted in figure 7. Because the FB feedback condition and FB no feedback condition only differed in procedure after children's approach in the first trial, and because there was no significant difference between performance in the two conditions (Fisher's exact test, $p = 0.640$), we collapsed them for the first trial analyses. In the first trial, in the FB condition, 18 of the 24 children (75%) incorrectly chose the referred box, which was significantly different from chance (binomial test, $p = 0.023$). In the TB condition, 10 of the 12 children (83%) correctly chose the referred box, which was significantly different from chance (binomial test, $p = 0.039$). There was no difference between the FB and the TB condition in children's choice of the referred box (Fisher's exact test, $p = 0.691$). There was no effect of sex in each condition (Fisher's exact tests, all $ps \geq 0.470$).

#### 3.2.2.2. Repeated trials
Thirty-four participants took all three trials, one took two trials (FB no feedback), and one took only one trial (FB feedback). Within each condition (TB, FB feedback, FB no feedback), there was no difference in performance between the three test trials (McNemar tests, all $ps = 1$). Within each test trial, there was no difference in performance between the three conditions (Fisher's exact tests, all $ps \geq 0.214$).

### 3.3. Discussion

In studies 2a and 2b, we did not find any evidence that 2-year-olds take an agent's belief into account when interpreting her referential communicative act. In all three task versions tested across studies 1 and 2, children predominantly chose the referred box, independent of the agent's TB or FB. In the modified versions of studies 2a and 2b, both aimed at reducing pragmatic ambiguities, we did not find that children adjusted their behaviour according to the agent's belief. Furthermore, we did not find any learning effects or enhanced performance when children received multiple trials with or without feedback. But it is possible that we did not find evidence for toddlers' FB ascription abilities in our modified versions, because we added not only more pragmatic context to the Sefo task, but also other extraneous and potentially detrimental task demands. However, because children chose the

referred box in all tasks, including the original task, and did not show any differences across versions it seems unlikely, that the additional factors of studies 2a and 2b (i.e. referential salience of the Sefo; and the agent's reason to request) selectively led to the low performance.

# 4. Study 3: validation

Studies 1 and 2 failed to find any evidence that children keep track of other agents' beliefs in communicative interactions in the Sefo task before the age at which they come to solve standard explicit FB tasks. The Sefo task may thus not be a reliable measure for precocious FB understanding in younger children. Nevertheless, is it a valid task that measures belief ascription at all? This is a pressing question because so far, the interpretation of the validity of the tasks rests merely on naive task analysis: when we consider and describe the interaction in the Sefo task, we find it obvious that it requires belief-tracking. But this may in fact not be so obvious to children, perhaps even adults (for similar questions regarding violation-of-expectation studies, see [41]). In other words, what we need beyond our intuitions regarding face validity is convergent validation: does performance in this task converge (correlate) with performance in established tasks?

The rationale of study 3 was, therefore, to address the convergent validity of the Sefo task in relation to standard explicit ToM tasks in populations that are known to have (some) explicit ToM abilities. Therefore, we designed two new explicit Sefo task versions, whose basic procedure was the same as the original task, with explicit test questions modelled after a recent study [32]. That is, we implemented a Label test question (Which one is the Sefo?) and a belief test question (Which one does she think is in the box?) instead of the interactive request. All participants received the original Sefo task, the two new explicit task versions, and standard explicit tasks. In study 3a, we tested 3;6–4;6-year-old children with three Sefo task versions and an adapted explicit change-of-location task. In study 3b, we tested adults with a pen-and-paper version of the original Sefo task and the two explicit versions.

## 4.1. Study 3a

The aim of study 3a was to validate the Sefo task, by testing older children (average age: 4 years) with the original task, two explicit task versions, and an adjusted explicit change-of-location task.

### 4.1.1. Methods

#### 4.1.1.1. Participants
The final sample consisted of twenty-four 3;6–4;6-year-old children (median age = 47;9 months, age range = 42–54 months, 12 girls) from mixed socio-economic background. Children were recruited from a database of children whose parents had previously given consent to experimental participation and tested in the facilities of the University of Göttingen by two female experimenters. In total, six children were tested but excluded owing to refusal of participation ($n = 5$), or experimental error ($n = 1$).

#### 4.1.1.2. Material
We presented three pairs of novel objects to the children (figure 8, order counterbalanced) and three pairs of boxes (one pair same-coloured, two pairs of different-coloured boxes that were counterbalanced), whose lids were attached to the top, so they remained in an upward position when opened. In the end, every child was tested with an explicit FB task as described in study 1. To keep the task types as parallel as possible, we also implemented a second object to the story.

#### 4.1.1.3. Design and procedure
All children were tested in a FB condition and received four trials in total. The original interactive Sefo task was always the first task being tested. The order of explicit Sefo tasks (label question and belief question) was counterbalanced. The explicit FB task was always the last task of each test session.

*4.1.1.3.1. Sefo tasks.* The test sessions always started with the original familiarization phase as described in study 1. After that, every child received three Sefo trials, with the first being always the original task with the interactive measurement of handing over one of the objects. Here, two same-coloured boxes were used. In half of the test sessions, E2 was absent during the object's placement, only appeared in the absence of E1,

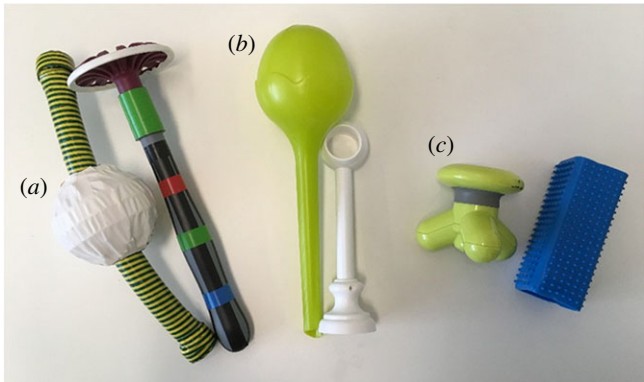

**Figure 8.** Pairs of objects used in the interactive and explicit Sefo tasks in study 3a ((*a*) a masher for potatoes and a ball prepared with a tube, (*b*) a plant watering bulb and a curtain holder, (*c*) a lint brush and a massage device).

and disappeared after the swap of the objects. After the test sequence in which the child handed over one of the objects, E1 stated that she was waiting for her friend to play with them. E2 appeared to E1's surprise from behind the curtain, stated that she wants to play with E1 and the child, and sat down next to the child. In the other half of test sessions, E2 was present from the beginning onwards. In each of the two following trials, we used two novel pairs of objects, two pairs of different-coloured boxes, and explicit test questions at the end instead of the interactive test sequence. The explicit trials followed the same basic procedure until the reference phase. E1 stopped after she pointed to one of the boxes while saying, 'Do you know what is in here? There is a Sefo in this box! A Sefo!' (in the second trial, the label 'Toma' was used instead). E1 then turned her back towards the child and E2. While E1 was not attentive, E2 placed laminated pictures of the two objects in front of the child (sides counterbalanced) and asked either: 'Which one of those two is the Sefo? (Toma)' (label question) or 'Which one of those two does [name of E1] think is in the [coloured] box?' (belief question). The order of test questions was counterbalanced. After the child pointed to one of the depicted objects, three control questions were asked: (i) 'Which one of those two did she put in the [coloured] box?', (ii) 'Which one of those two is now in the [coloured] box?' and (iii) 'And who put it there?'. The following trial started with E1 getting two novel objects and removing the still closed boxes so that the outcome was never presented to the child.

*4.1.1.3.2. Explicit false belief task.* After children received three Sefo tasks, the explicit FB task was introduced as an additional story and followed the same procedure as described in study 1, except that we introduced a second object to keep tasks as parallel as possible within age group. The second object was introduced after the protagonist had played and expressed her obsession with the first object to emphasize the object of focus. Before the protagonist left the scene, she placed each object in one box. The second protagonist swapped the objects exactly as in the Sefo task: she took the first object out of the box and placed it in front of the box, took the second object out of the box, showed it to the child, placed it in the first box; took the first object, showed it to the child and placed it in the second box and disappeared again. With the return of the protagonist, the test question (When Luchsi returns, where will Luchsi look for his car first?) was asked. After children answered the test question, three control questions were asked ((i) 'Where did Luchsi put his car in the beginning?', (ii) 'Where is the car now?' and (iii) 'And who put it there?').

### 4.1.2. Results

In the original Sefo task, all 24 children chose the referred box (binomial test, $p = 1$; figure 9). In both explicit Sefo trials, children performed at chance level (binomial tests, label question: $p = 0.15$, belief question: $p = 0.84$). Performance in both explicit tasks differed significantly from the original task (McNemar tests, $ps < 0.001$), but no significant difference between the two explicit tasks was found (McNemar test, $p = 0.55$). In the explicit false belief task, children performed at chance level (binomial test, $p = 0.54$) and there were no differences to the explicit Sefo tasks (McNemar tests, label question: $p = 0.21$, belief question: $p = 0.58$). No correlations between tasks were found (table 6). We neither found order effects nor an effect of E2's position during the original Sefo task (Fisher's exact test, all $ps > 0.68$).

In addition, we checked for developmental convergence for the interactive Sefo task within the group of 3-year-olds. Therefore, we merged data of children that were 36–48 months old and participated in the FB condition of either study 1 (24 participants) or study 3a (11 participants). Within the age group

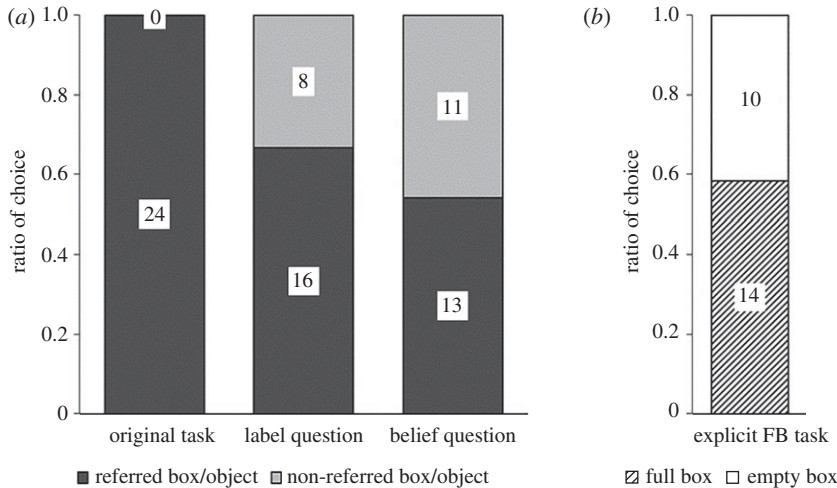

**Figure 9.** Proportion of participants who chose each box or depicted object in (*a*) the different Sefo-task versions and (*b*) the explicit FB task. Numbers in bars show number of participants.

**Table 6.** $\phi$-correlations between the different explicit tasks in study 3a.

|                | belief question | explicit FB task |
|----------------|-----------------|------------------|
| label question | 0.06            | −0.3             |
| belief question |                | −0.07            |

of 3-year-olds, 27 children chose the referred box while eight children chose the non-referred box. This performance differed significantly from chance (binomial test, $p = 0.05$).

## 4.2. Study 3b

The aim of study 3b was to validate the Sefo task and the explicit task versions with a survey on adult participants. Therefore, we put the interactive request of the original Sefo task into a question ('Which of those two things would you hand her over?' (original task question)) and asked the same explicit test questions as in study 3a (label question: 'Which one is the Sefo?' and belief question: 'Which one does she think is in the box?').

### 4.2.1. Methods

#### 4.2.1.1. Participants
Forty-eight adults (median age = 22 years, age range = 18–34 years; 21 males, 26, females, one no answer) were recruited and tested on the main campus of the University of Göttingen.

#### 4.2.1.2. Design and procedure
In total, adults received three pen-and-paper versions of the Sefo task with the original task being the first task tested. The order of the two explicit Sefo tasks (label task and belief task) was counterbalanced. Before adults answered the pen-and-paper survey, a video of the Sefo task from the child's perspective was shown (videos can be found at https://osf.io/yvtx2). In the video, an unrecognizable masher for potatoes and a ball prepared with a tube (figure 8, object pair *a*)) were used with counterbalanced sides of placement. E1's pointing side was also counterbalanced. The survey included one question resembling the request in the original Sefo task ('Which one of those two objects would you hand her over?' (original task question)) and two counterbalanced explicit test questions ('Which one of those two is the Sefo?' (label question), 'Which one of those two does [name of E1] think is in the box she pointed at?' (belief question)).

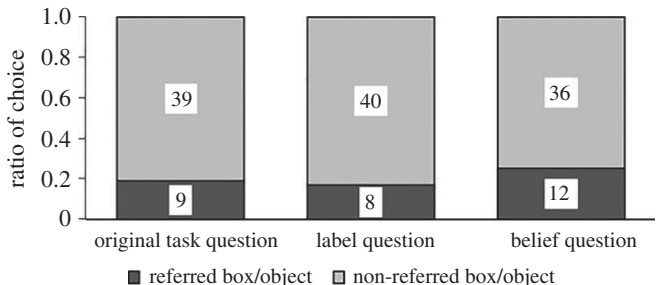

**Figure 10.** Proportion of participants who chose each box or depicted object in the three test questions for adults, study 3b. Numbers in bars show number of participants.

**Table 7.** $\phi$-correlations between the different explicit tasks in study 3b. (* $p < 0.01$; ** $p \leq 0.001$.)

|  | label question | belief question |
| --- | --- | --- |
| original task question | 0.71** | 0.47** |
| label question |  | 0.42* |

### 4.2.2. Results

Adults' answers in all three test questions differed significantly from chance (binomial tests, all $p$s $< 0.001$, figure 10). That is, they predominantly chose the non-referred box/object in all tasks. No significant differences between the questions were found (Fisher's exact test, $p = 0.31$). Correlations between the questions are shown in table 7.

### 4.3. Discussion

The main findings of studies 3a and 3b were the following: first, study 3a failed to reproduce the FB pattern of the original Sefo task in older children. In the interactive task version, none of the children considered E1's belief when responding to her request. Second, in the explicit task versions, in which children were asked about the objects' labels and E1's belief about the object's location, and in the explicit FB task they chose at chance level. Third, there were no correlations between the explicit Sefo task versions and the explicit FB task. Fourth, adults, however, were able to identify the object E1 was referring to in all three task versions and their performance correlated across tasks. In summary, thus, it seems that the Sefo task does measure what it is supposed to measure (belief ascription), at least in adults. Regarding children, however, it remains somewhat unclear what exactly the interactive task version taps and whether it is really more sensitive and less demanding when it comes to FB understanding, compared to explicit versions. One possible explanation for the older children's performance in the interactive version may be the high demands on inhibitory control during the experimenter's request. Recent research showed that 3- to 4-year-old children generally find it very difficult not to follow an adult's pointing gesture and, for example, do not differentiate between an ignorant and a knowledgeable experimenter when it comes to the interpretation of a pointing gesture [42]. In the present context, thus, older children could fail the Sefo task by simply following the experimenter's pointing to the referred box.

## 5. General discussion

Given that the Sefo task is the most influential and unambiguous interactive task that has produced evidence for sophisticated ToM (belief ascription) in infants, with far-reaching theoretical implications, and given recent difficulties with replicating diverse implicit FB tasks, the aim of the present study was to systematically investigate the reliability and validity of this task. The main findings of the present studies, conducted in collaboration across two laboratories, were the following: study 1 failed to reproduce the original results in 17-month-old infants (original age group), and also in 3-year-olds. Additionally, we

found no correlation between the Sefo task and the standard FB task in the older age group. In study 2, we tested 2-year-olds with the original task and two modified versions, aiming at increasing pragmatic plausibility, and failed to replicate the original pattern of results. The majority of children chose the box the experimenter referred to, irrespective of the experimenter's belief. Study 3 did not find any evidence that older children pass the FB condition of the Sefo task. Even though 3;6–4;6-year-old children performed at chance level in explicit task versions, all children retrieved the object out of the box E1 was referring to in the original interactive task. Again, we did not find correlations with a standard explicit FB task. Only adults were able to interpret E1's referential communication appropriately and identified the correct object in the original tasks and the explicit versions, and correlations between the tasks were found. Furthermore, we found evidence for a counterintuitive, but recurring pattern of children around the age of 4 failing in standard TB tasks (e.g. [36,38]). Recent research showed that pragmatic performance factors lead to confusion in this specific trivial test situation and thus to incorrect answers [39,43].

The emerging picture is thus the following: none of the age groups of children between 17 months and 4 years tested here reliably identified the object the protagonist was referring to when she held a false belief. Seventeen-month- and 3-year-olds chose at chance level, 2-year-old and 4-year-old children chose the referred box predominantly. While young infants' chance performance in both conditions might be explained by extraneous performance factors and procedural ambiguity, 3-year-olds chance performance in study 1 might appear inconsistent with this, because 2- and 4-year-olds clearly failed the task. However, regarding our additional analyses that focused on 3-year-old participants from studies 1 and 3a, it seems likely that also at age 3, children predominantly choose the referred box and fail the FB condition of the Sefo task. Thus, our findings are in line with other recent replication studies that failed to replicate the original results [18,34]. In principle, in adults, it seems that the Sefo task taps what it was designed to tap, namely belief-tracking, albeit on an explicit level. Nevertheless, the present findings suggest that the Sefo task may not measure *early* FB ascription abilities, even though it was designed to uncover precocious FB understanding in more sensitive ways than standard explicit tasks.

Why do the present findings differ from the original ones of Southgate *et al.* [12]? There are two broad possibilities. First, young children may indeed operate with precocious meta-representational belief understanding that appropriate interactive methods are able to uncover. The present studies may then differ from the original ones in subtle but crucial ways in failing to implement these appropriate methods, for example in the interaction style of the experimenters (with corresponding consequences for children's engagement during the test situation). Maybe, for example, our experimenters did not use ostensive communication as proficiently as in the original study. Whether there were such differences and whether these would make a difference, though, cannot be answered in the current study. Note, however, that, of course, great care was taken to ensure children's engagement and rapport with the experimenters. Importantly, our studies were carried out in collaboration between two laboratories, and in each laboratory we assigned different experimenters between studies. Yet, none of the different experimenters was successful in uncovering children's precocious FB understanding, and no differences between laboratories were found. Another difference might be the phrasing of the experimenter's request. In study 1, we used the first part of the phrasing of Southgate *et al.*'s exp. 3 (Do you know what's in here?), but did not use the novel label (Sefo). Is it possible that infants failed because of the missing chance to learn a new object label? This explanation appears unlikely, because infants in the original study performed similarly with or without word learning, and in light of a recent study by Papafragou *et al.* [32] that casts doubt on the performance-enhancing effect of word learning contexts in FB tasks *per se*. Additionally, in our studies 2a and 3a, we used the exact phrasing of Southgate *et al.*'s exp. 3 including the novel label, but still failed (see also Dörrenberg *et al.* [18] that even failed with the phrasing of Southgate *et al.*'s exp. 1).

The second possibility regarding the discrepancy between our findings and the original ones is that the original study (in concert with publication bias, file-drawer problems and the like) was based on false positives, or at least overestimated the actual effect size. Only large-scale, collaborative replication projects may be able to demonstrate whether this is the case.

However, even more dramatically and somewhat ironically, we did not only fail to reproduce the original findings but found that the Sefo task may actually be *less* sensitive than standard tasks and actually *under*estimate young children's belief understanding. Empirically, this is at least what the results of 3- to 4;6-year-old children in studies 1 and 3a seem to suggest in which the performance was actually less poor in explicit tasks than in the interactive Sefo task. Conceptually, such a possibility (that the interactive Sefo tasks may in fact be particularly difficult) seems highly plausible considering its task demands and in the light of recent findings: what the child needs to do in the Sefo FB condition is to follow the agent's pointing towards box 1, and then suppress the tendency to choose this box, but rather

go for the other one. This raises massive task demands in terms of inhibitory control of a prepotent response (going with the point). We know from much recent research that such inhibition is indeed cognitively taxing and develops in protracted ways over the preschool years: for example, 3- to 4-year-olds have been found to exhibit a robust bias to search in pointed-to locations even when the pointer has turned out to be unreliable, deceptive or ignorant about the object location (e.g. [42,44–47]). In a study by Palmquist & Jaswal [42], for instance, children saw a video in which one actor hid an object under one of two cups, while another actor covered her eyes and did not see the hiding location. The hider used a barrier so that also the child was ignorant. After the hiding, children were asked who of the two actors knew where the object was. When both actors sat with their hands in their lap during the test question or each grasped the top of a different cup, children significantly selected the actor that hid the object. When both actors pointed at different cups, though, children selected at chance level and no longer discriminated between the knowledgeable and the ignorant actor. This indicates that pointing may interfere with children's cognitive capacities, leading them to over-attribute knowledge also to obviously ignorant agents. Accordingly, even if children would understand the false belief of the agent in the Sefo task, they may still fail owing to problems in inhibiting their bias to search in pointed-to locations. In addition, linguistic demands may also be increased in the Sefo task. For instance, the request of the experimenter elicits a choice between the two boxes which both contain an interesting toy. An unambiguous understanding of that request requires not only interpreting the pointing gesture but also the verbal prompt. Thus, passing the Sefo task may not only require ToM as such, but also advanced inhibition, decision-making and language skills—all of which develop in protracted ways in early childhood. Rather than being the solution to false negative results from explicit tasks, as initially intended, the Sefo task may itself produce false negative findings vis-à-vis traditional explicit tasks.

In future research, the reliability and validity of the Sefo and related interactive FB tasks need to be investigated in systematic and in-depth ways, ideally in a multi-laboratory, pre-registered collaborative endeavour such as ManyBabies [48].

Ethics. The studies were approved by the University of Göttingen ethics committee. All procedures performed in the reported experiments were in accordance with the 1964 Helsinki declaration and its later amendments or comparable ethical standards.

Data accessibility. The data generated in this study are available at https://osf.io/yvtx2.

Authors' contributions. L.W. and S.D. were involved in the design of the work, acquisition of data, analyses, and drafting the manuscript. M.P. was involved in the design of the work. U.L. and H.R. were involved in the conception and design of the work and supervising the data analysis. All authors interpreted the data, prepared the manuscript and approved the final version before submission.

Competing interests. We declare we have no competing interests.

Funding. This study was supported by the German Research Foundation (LI 1989/3-1, RA 2155/4-1, Project: FOR 2253). We further acknowledge support by the Open Access Publications Funds of the Göttingen University.

Acknowledgements. We would like to thank Annika Braun, Leslie-Ann Eickhoff, Rocío Fernandez, Anna Fink, Marc Heuser, Tijana Lajic, Joana Lonquich, Laura Meier, Senta von Münchow, Rieke Oesterreich and Julia Ruge for help with data collection, coding and recruiting participants, as well as Marlen Kaufmann, Konstanze Schirmer and Jessica Schröter for laboratory coordination. We are grateful for the participation of all the parents and children.

# Appendix A. Alternative measures of the original replication (Study 1)

In order to check for other potential indicators of children's belief ascription abilities, we analysed additional behavioural responses of the 17-month-olds. From the point of time when the boxes had been opened until their first movement towards one of them, we coded infants' latency in decision-making and the direction of their first look. Overall, infants took longer to make a decision in the TB condition (8344 ms, s.d. = 13 682) compared to the FB condition (2858 ms, s.d. = 2635). However, this difference was statistically not significant (Mann–Whitney $U$ = 207, $Z$ = −1.671, $p$ = 0.095). For further post-hoc analyses, we also coded the direction of infants' first look. Because we did not aim initially at analysing gaze behaviour, we were able to extract looking behaviour from 39 out of 48 videos (TB: 17, FB: 22). In the remaining nine videos, infants' eye gaze could not be coded reliably either because of the quality of the videos or the camera angle not facing the child. In both conditions infants' first look was equally often directed towards the referred box than to the non-referred box (Fisher's exact test, $p$ = 0.556). While 53% looked to the referred box in the TB condition (nine infants), 50% did so in the FB condition (11 infants; binomial tests, both $p$s = 1).

In line with the results presented in study 1, we did not find any differences between TB and FB conditions. Even after including alternative measures, we could not find further conclusive evidence that would support the ability of belief attribution in infants.

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
