## [Reviewer comments · Royal Society Open Science]

Review History

RSOS-191998.R0 (Original submission)

Review form: Reviewer 1

Is the manuscript scientifically sound in its present form?

Yes

Are the interpretations and conclusions justified by the results?

Yes

Is the language acceptable?

Yes

Do you have any ethical concerns with this paper?

No

Have you any concerns about statistical analyses in this paper?

No

Recommendation?

Accept with minor revision (please list in comments)

Comments to the Author(s)

This manuscript reports three studies, attempting to replicate and extend an interactive paradigm by Southgate et al. (2010) that was meant to tap into infants' and toddlers' theory of mind abilities. Results using a close approximation of the original study failed to replicate its main finding (that 17-month-olds seemed to understand an actor's false belief), and the additional studies with various manipulations to attempt to make the task easier/more pragmatically felicitous also failed to demonstrate that toddlers (and even 3- and 4-year-olds) would respond in accord with an understanding of false belief. Adults, however, responded as expected on the "Sefo" task and their performance was correlated with other theory of mind measures.

This is a well-written paper, reporting a programmatic series of studies. I appreciated that the authors went beyond demonstrating a "failure to replicate" by attempting to modify the task still further and to investigate e.g., its relation to other tasks. I have a few comments and suggestions.

1. My primary comment is that I believe the authors have demonstrated conclusively that they were unable to replicate findings from the original task. But this leaves open the question (which I don't think the authors address, though I may have missed it) about why the discrepancy? Is it because of the small sample size in the original study? Were the authors able to confirm with the original researchers that the procedure was done in exactly the same way? As I know the authors are aware, failures to replicate can occur for many reasons and it would be good to be able to work with the original researchers to try to understand together why their findings were different.

2. I would also encourage the authors to consider further the implications of their findings. They argue persuasively that whether toddlers understand false beliefs has important theoretical implications re nativism, for example. This seems reasonable. But it seems like overstating the case to suggest that theory of mind reduces to an understanding of false beliefs, which I don't think the authors want to argue but which is an interpretation a reader could make. Additionally, I'd encourage the authors to try to link the abilities they believe toddlers have or do not have to children's real-world experience. What would it mean (beyond performance on experimental tasks or theoretical debates about nativism) for children to have the kind of understanding that some scientists attribute to them?

3. I wondered whether the authors could code from video other aspects of the children's performance that could also speak to some of the debates in the field. For example, were children slower to make responses in the TB than FB conditions (even though they ended up being at chance in which object they chose)? Did they look more at one than the other object/box depending on condition? What kind of spontaneous comments (if any) did children make – did they request clarification? What kinds of individual differences might obtain in these kinds of tasks?

Review form: Reviewer 2

Is the manuscript scientifically sound in its present form?

No

Are the interpretations and conclusions justified by the results?

No

Is the language acceptable?

Yes

Do you have any ethical concerns with this paper?

No

Have you any concerns about statistical analyses in this paper?

No

Recommendation?

Major revision is needed (please make suggestions in comments)

Comments to the Author(s)

Review of RSOS-191998

Entitled 'Actions do not speak louder than words in an interactive false belief task'

Overview:

This paper presents 3 experiments with the expressed objective to replicate (closely and with modifications) the study of Southgate, Chevallier and Csibra, 2010. The paper mainly focuses on the relevance of the replication approach and on the difficulties in finding robust and reliable methods for studying the development of theory of mind. No doubt, replicability is a central issue for research lead by the idea of accumulating valid knowledge, and psychology is an area of research facing a serious replication crisis. That said, it is extremely important to pursue transparent and stringent research practices in such replication attempts as well and at the same time the theoretical framing should remain objective.

I appreciate this general approach together with the efforts the authors invested into empirical testing, nonetheless, I have concerns with respect to their presentation of data and the conclusions that can be drawn from them. I miss important details in the presented methods and results and find the interpretation of the selected pattern of data somewhat biased.

In order to this paper become more stringent, persuasive and constructive, presentation of additional methodological details, data and a more toned down interpretation is proposed.

Concerns in details:

Introduction:

My main impression is that the authors handle replication problems and theory driven debates on the interpretation of given findings as the two facets of the same problem (the authors handle these as reliability and validity of the empirical phenomenon). While both questions have the highest importance, and clearly they are related, these issues cannot be directly equated. The entire introduction suffers from this latter issue (the validity of a phenomenon is not always handled correctly in my opinion). Bringing up replication attempts and their results as the main focus of the paper seems to justify the selection to include only those studies that has been targeted to be replicated. Clearly, this approach results in a selective set of references. This is fine unless someone does want to question the underlying theoretical constructs as well, in which case the related conceptual approaches (though not necessarily being replications) would be important to be mentioned and considered in details. Otherwise the reader receives a biased picture of the state of the art research in the field. So, if the authors' objectives entail the discussion of theoretical approaches as well, they ought to include the contrastive explanatory angle and the empirical bases of them too.

Let me highlight this problem with the help of some examples:

1. One of the the authors' main claims is:

'In fact, a growing body of replication studies puts into question the robustness of the original evidence on early and automatic ToM (for an overview, see Kulke & Rakoczy, 2018).' p. 5

Following this claim, the authors mention the failed replications of both anticipatory looking tasks and looking time studies, however they ignore the commentary that focuses on the problems with the above replication attempts, see Baillargeon, Buttelmann, Southgate, 2018. Also, the authors do not mention that a collection of conceptual replications are reviewed in the

literature (see Scott and Baillargeon, 2017). The purpose of representing the debate itself - and not only their own take on - would necessitate discussing the other viewpoint, or at least referring to it.

2. Here is another example on what kind of phrases and claims I find misleading: ‘Similarly, ToM-based priming effects in implicit tasks with adults could recently also not be replicated (Conway, Lee, Ojaghi, Catmur & Bird, 2017).’ p. 5.

This sentence suggests that implicit ToM-based priming tasks in general are not replicable. There are several problems with this claim then. First, a closer look at the referred paper reveals that the objective of the study is to offer a competing interpretation for an empirically reliable (replicable) phenomenon. So, while providing a conceptual replication of the phenomenon, the paper aims to disentangle different explanations beyond it. That is true, the findings seem to support a non-mentalistic interpretation of the data, yet this study is NOT an exemplar of a non-replication. Second, even if the role of the study in the paper was to represent problems with replicating similar studies, the authors should not dismiss that the actual phenomenon has been replicated by couple of other studies conceptually, with contrasting interpretation to the one mentioned (see Furlanetto et al., 2016; Marshall et al. 2018; Wiese et al., 2012; Teufel et al., 2010), and there are other paradigms on perspective taking with similar theoretical background that are actually highly replicable across laboratories (Surtees et al., 2016; Elekes et al., 2016.)

3. With respect to interactive FB tasks, the manuscript focuses on the so called helping task by Buttelmann et al., 2009 and the attempts to replicate it. The argument in the manuscript is that while this empirical paradigm is at least partially replicable, there is a problem with the validity of the original task, as it ‘may not measure belief-tracking but simpler forms of social cognition such as goal-understanding.’ p. 6

I am fine with the claim that the responses given in the experimental protocol can be interpreted differently, however the dispute would be mentioned in more details, to make obvious for the readers in what sense the validity of the protocol is in question. I am emphasizing this as the alternative interpretation itself also received criticism, namely that the teleological approach mentioned can explain how children bring up reasons for an already performed behavior (not necessarily ‘a simpler form of social cognition’), but based on that approach there is no answer on how and whether children are able to predict the behavior of a protagonist -the original problem of ToM research (see <http://cognitionandculture.net/blogs/pierre-jacob/if-teleology-is-the-answer-what-was-the-question/>).

4. Arriving to the focal purpose of the paper, the authors introduce the sefo task. I miss here the detailed description of the Loth and Happé task (2002) and also of the task of Carpenter et al (2002). Though done with 3 year olds, those studies served as sources for the sefo task and actually provide convergent evidence that 3 year-olds are able to exploit belief ascription for the sake of word learning.

5. In relation to the sefo task the already existing replication attempts are discussed. Here, I would like to suggest a careful and thorough description and interpretation of data in general, as this is the most helpful strategy to support open science practices and the understanding of the robustness of the phenomenon.

So, with respect to the Grosse Wiesmann et al. (2017) study, it would be important to mention that the main question in such a task is whether the response pattern in the FB condition shifts away from children’s natural (baseline) response to a request accompanied by a pointing gesture which is measured in the TB condition. Therefore, a comparison between TB and FB performance is needed in order to grasp the modulatory effect of belief ascription in the interaction, that is actually missing in the above experiment (while that is true that the original study could also show above-chance performance for the FB conditions).

With respect to the Kiraly et al (2018) study the authors raise: ‘a main concern is that this finding is difficult to interpret given that no analogous prospective TB condition was conducted.’ I do not

see the main concern here. Actually, there was a TB condition that was used as a bases for comparison for the prospective FB condition as well: at the beginning of the protocol infants explored a pair of sunglasses that were see through, and then they saw the model using an other pair of sunglasses. True, that in this condition children got themselves information that those pair of sunglasses the model wore were transparent as well. Is there any theory that would predict difference in performance just based on receiving confirmatory information later? In what sense a 'just prospective TB condition would modify the results? So, this concern is of no base, it is simply misleading.

Clearly true that this conceptual replication contained modifications that were difficult for young children in the specific TB-FB condition in terms of memory load, where 18-mo-olds failed, but that was the target question of the paper of Kiraly et al. (children had to keep track of the status of the sunglasses and relate that to the possible belief content of the protagonist with the help of their memory capacities). What is the relevance then of the description of this modification in the present paper without mentioning the target question of the original paper? Again, this is somewhat misleading, as it is presented as a concern.

In addition, both for the Grosse Wiesmann study and the Kiraly et al. study the argument is brought up that the absence of the deceptive motive could have led to the low performance in the FB condition. Is it the claim that just based on the fact that the model remained in the room during the test in both studies the deceptive mode is questionable? In my opinion, as actually the deceptive mode was performed in both studies while the model was present, the authors should also try to give an explanation on how children interpret the deceptive mode in the presence of the model. So, the suggestion is either give more details on this explanation or leave out the post hoc interpretation, as it cannot give a systematic explanation (same context resulted in failure in 3- and 4-year-olds in Grosse Wiesmann et al study, failure in 18-mo-olds in Kiraly et al study, yet in success in 3-year-olds in Kiraly et al study).

The concerns with respect to the linguistic complexity used for older children are also somewhat confusing. If I understand correctly, the argument is that in case of the study of Kiraly et al., the use of the original phrases from Experiment 1 of Southgate et al. (Do you remember what I put here?) would have induced a pragmatic shortcut for 3-year-olds to solve the task without belief ascription. But only for the FB condition, since, according to the authors: 'due to different pragmatic contexts between conditions, children may interpret the prompt in the TB condition differently'. What are those differences in the pragmatic context (for the TB and FB condition) that are completely unrelated from belief ascription? Please describe the details, I do not find them obvious. Rather, this claim would predict pragmatic ambiguity for the TB condition as well if children do not take into account the belief of the speaker. (Note again, the original study did show the same pattern of results with three different prompting).

Study 1: Direct replication

Methodological issues

1. I find compelling that '40% of the 17-months-olds tested in Study 1 struggled to pass the original familiarization criterion'. p 16. The authors therefore decided to a) modify the procedure slightly for some participants, and b) applied a more lenient passing criteria for a subsample of participants. One can find a sample description in the result section, referring to the lenient criteria passers: FB: n = 12, TB: n = 8. So then, the sample of children who passed the original criterion (FB: n = 12, TB: n = 16) includes those participants for whom the procedure was modified slightly or not? This particular detail could be of importance to evaluate the discussion of the difficulties of familiarization in relation to the reliability of the phenomenon. The authors argue for the reliability of the protocol by emphasizing that there was no performance difference between the subsamples that did vs. did not reach the familiarization criterion. This claim would receive more emphasis if the subsample of passers would not include those who received modified demonstrations.

(The claim that for 3-year-olds the exact same familiarization did not provide any challenge while they also failed in the test is inconclusive already because of the potential age differences: the fact

that there was no problem for older children does not rule out that there could have been issues for younger ones. Nonetheless, actually there was a tendency difference in the performance in TB vs FB conditions in the older age-group).

2. There are some missing minor methodological details by the clarification of which the methods would be more clear. When did E2 introduced the deceptive mode? Already at the beginning of the location swap episode? Was this deception mode there for the TB condition as well? How long was E1 away? How many pointing were shown in the test phase (just one or more)?

3. While the authors explicitly declare that Study 1 is a direct and as close as possible replication of the original study they modify the verbal prompts used in the task. The verbal prompts of Experiment 2 and Experiment 3 of the original study were combined. The result is: "Do you know what's in here? I want to play with this" ... "Can you give it to me?". (Just for comparison, verbal prompts of the original study: Exp 2 of Southgate et al.: Do you remember what I put in here? Shall we play with it? Shall we play with it? Let's play with it! Finally, she said, 'Can you get it for me?' Exp 3: Do you know what's in here? There's a sefo in the box! There's a sefo in here! Can you get it for me?)

The authors themselves call attention to the potential effect of the pragmatic ambiguity of some verbal prompts that can overshadow children's belief ascription capabilities, (likewise they argue for the case of the prompt used by Grosse Wiessman et al, 2017; see page 8.) I myself think that while the purpose of this new prompt could have been to use the most neutral phrasing, the result of the choice is again equivocality. The assumption beyond the interactive FB task is that word label mapping or reference mapping trigger belief ascription: in search for a correct mapping of the referent of the interlocutor's request the child need to track the belief state of her. However, what is the message of the prompts used in this study? The model asks whether the CHILD knows what is in the box, whatever it is, and clearly, she expresses she would like to play with that object. In contrast, the versions of the original study highlight the possible mismatch between the protagonist's perspective and the child's knowledge, either by referring to a previous event and a previous state of the scenario (using the phrase 'Do you remember what I put in here?') or by introducing a novel label, for the mapping of which the child needs to track the protagonist mental states (she cannot know that the object in here is the SEFO). So, the pragmatic call of the phrase in the present version is different from that of the originals. I think it is highly relevant to acknowledge this difference in the paper. Just saying that this prompt is similar to the original version of Exp 3 is inaccurate. Based on this, it is questionable to call the study as direct replication.

4. The study introduced three consecutive trials for testing 17-mo-olds where feedback was given. The text describes the forms of feedback for the FB conditions, but what was the feedback for the TB conditions?

5. It is unclear from the text that in the case of the 3-year-old group each child was tested in only TB conditions or FB conditions? That is what the following sentence means? „3 -year-olds were tested with a single Sefo FB/TB test trial and one trial of a corresponding standard FB/TB (change-of-location) task (task order counterbalanced).“

Results

1. What is the exact reason for collapsing the sample and check whether there is a tendency to choose the referred box?

2. Can the authors report a matrix of failers/passers with respect to the sefo and the standard task? If I understand correctly, this data is available, though only separately for TB and FB conditions.

3. I doubt that a correlation analysis is well grounded when the data for the to-be-correlated measures on both sides show random pattern!! At least, no correlation in such case is definitely inconclusive.

Discussion

1. Is there any explanation on why there was no learning effect in the TB condition for 17-month-olds?
2. What does the standard TB version measure?
3. In discussion of the performance of the 3-year-olds, (who actually performed at chance in both standard FB and TB tasks), instead of accepting that children were simply guessing in both kinds of tasks, the interpretation is raised that there are two separate subsamples (FB passers/TB non-passers and TB passers/FB non-passers in the group of 3-year-olds. This interpretation sounds very speculative. Is there any solution to prove it based on the data? Otherwise, based on Occam's razor, (and also based on the data on TB non-passers introduced in the paper they cite) it is more likely that the study represents a no replication of the TB version of the classic task.

Study 2a

Methodological issues

1. Note, the pragmatic ambiguities revealed by the verbal prompts hold fold the 'original' sefo task here as well, as described above.
2. Actually, in the modified task the verbal prompting is different, it contains a new verbal labelling for one of the new objects as in Exp3 of Southgate et al. However, based on the description, the experimenter continuously pointed at the box during the verbal prompting. This is again a difference from the original study. Was there a difference in this respect between Study 1 and the 'original' trials of Study 2a? Worth to mention this detail, as this strong referential cue can overwrite children's knowledge based inferences (as the authors themselves discuss this possibility in the general discussion, but not as part of giving the rationale for Study 2b).
3. Were there any feedback or related verbal description between the trials?

Results

Clearly, in all conditions children followed the pointing behaviour of the experimenter and chose the object from the referred box. I think that there is no reason to calculate correlation in case of such homogenous pattern at ceiling.

Study 2b

I find the objective and the implementation of this task very interesting. Ecologically it seems to be more valid, the request of the experimenter emerges from an obvious need. However, I still have doubts whether this implementation is pragmatically more transparent than the previous ones. In my reading, the familiarization of the task assures children that the experimenter knows what makes an apparatus work. When the experimenter is back and deliberately referring to one of two boxes -each of which contains a novel object -, and says 'We need what is in there. Can you give it to me?', this can be taken as an assertion that they (the child and the experimenter) need the specific object that is in the box the experimenter actually is referring to. As previously she turned out to be knowledgeable, no doubt emerges in the situation.

I would like to underline again that in previous versions a tension between the referential cuing and the assertion was present ('remember' or the need to map a novel label to an object).

Study 3

The question remains open in what context infants literally follow the strong referential cuing and when they are sensitive to the belief of the model. Study 3 proposes to investigate this

question. The general introductory argument is that there was a repetitive failure in finding evidence that the sefo task is measuring whether young children keep track of other agents' beliefs in communicative interactions. The theoretical perspective can be formed differently: introducing modifications to the sefo task may alter the interpretation of the ongoing interaction, and consequently, the interpretation of the communicative intention of the partner by infants: in most of the cases they are led by the teaching intent of the protagonist, when it occurs within highly ostensive cues. The discussion and introduction would benefit from raising this alternative (Csibra and Gergely, 2009).

There is no clear description on what the explicit question versions of the Sefo task intended to measure. Indeed, they offer a possibility to answer the question whether children reacted differently in the behavioural request situation and in the label and belief questions. As those answers are coming from the same participants, a related sample analysis is suggested: that could show that there were differences in the answer patterns for the different tasks, thus potentially different interpretation of the request and the questions on the part of the same participants. I think the McNemar test fails to grasp the change between choices within participants.

Still, there is no discussion on the fact that the same children intended to follow directly the referential request of the experimenter, while their belief mapping and labelling attempts were significantly different from their behavioural choice. What is the explanation for this?

General discussion

I completely agree with the authors that the Sefo task in the future need to be more systematically investigated and tested in multilab collaboration. However, before arriving to this conclusion, it would be worth to summarize that even in the present paper there were some inconsistencies in result patterns (chance level performance for 17-mo olds for both FB and TB sefo conditions in Study 1, tendency difference for 3 year olds between FB and TB conditions in Study 1; always choosing the referred box irrespective of condition at ceiling in Study 2; difference in the pattern of responses for behavioural request and the verbal questions in the sefo trials in Study 3) that highlights that the sefo task as being an interactive task, the focus of interaction can be shifted sensitively to contextual factors.

The problems with the standard TB task should also be mentioned in the general discussion.

The discussion of the factor that the pattern of data in Study 2 and Study 3a is similar to the tendency found in the literature that 3- to 4-year-olds exhibit a robust bias to search in pointed-to locations even when the pointer has turned out to be unreliable, deceptive or ignorant about the object location deepens the need to study further what influence the behavior of children in such interactive tasks.

References:

- Baillargeon, R., Buttelmann, D., Southgate, V. (2018). Invited Commentary: Interpreting failed replications of early falsebelief findings: Methodological and theoretical considerations. *Cognitive Development* 46. 112-124
- Csibra, G., Gergely, Gy. (2009) *Natural Pedagogy*. *Trends in Cognitive Sciences* 13(4):148-53
- Elekes, F., Varga, M., & Király, I. (2016). Evidence for spontaneous level-2 perspective taking in adults. *Consciousness and Cognition*, 41, 93-103.
- Furlanetto T, Becchio C, Samson D, Apperly I. (2016). Altercentric interference in level 1 visual perspective taking reflects the ascription of mental states, not submentalizing. *J Exp Psychol Hum Percept Perform*. 42(2):158-63.

Marshall, J., Gollwitzer, A., Santos, L.R. (2018). Does altercentric interference rely on mentalizing?: Results from two level-1 perspective-taking tasks. *PLoS One*, 13(2), e0194101. doi: 10.1371/journal.pone.0194101

Scott, R. M., & Baillargeon, R. (2017). Early false-belief understanding. *Trends in Cognitive Sciences*, 21(4), 237-249.

Surtees, A., Apperly, I., & Samson, D. (2016). I've got your number: Spontaneous perspective-taking in an interactive task. *Cognition*, 150, 43-52.

Teufel, C., Alexis, D. M., Clayton, N. S., & Davis, G. (2010). Mental-state attribution drives rapid, reflexive gaze following. *Attention, Perception & Psychophysics*, 72(3), 695-705.

Wiese, E., Wykowska, A., Zwickel, J., & Müller, H. J. (2012). I see what you mean: How attentional selection is shaped by ascribing intentions to others. *PLoS One*, 7(9), e45391.

Decision letter (RSOS-191998.R0)

26-Feb-2020

Dear Mrs Wenzel,

The editors assigned to your paper ("Actions do not speak louder than words in an interactive false belief task") have now received comments from reviewers. We would like you to revise your paper in accordance with the referee and Associate Editor suggestions which can be found below (not including confidential reports to the Editor). Please note this decision does not guarantee eventual acceptance.

Please submit a copy of your revised paper before 20-Mar-2020. Please note that the revision deadline will expire at 00.00am on this date. If we do not hear from you within this time then it will be assumed that the paper has been withdrawn. In exceptional circumstances, extensions may be possible if agreed with the Editorial Office in advance. We do not allow multiple rounds of revision so we urge you to make every effort to fully address all of the comments at this stage. If deemed necessary by the Editors, your manuscript will be sent back to one or more of the original reviewers for assessment. If the original reviewers are not available, we may invite new reviewers.

- Data accessibility

If you wish to submit your supporting data or code to Dryad (<http://datadryad.org/>), or modify your current submission to dryad, please use the following link:
<http://datadryad.org/submit?journalID=RSOS&manu=RSOS-191998>

- Competing interests

- Authors' contributions

- Acknowledgements

- Funding statement

on behalf of Dr Teodora Gliga (Associate Editor) and Essi Viding (Subject Editor)
 openscience@royalsociety.org

Associate Editor's comments (Dr Teodora Gliga):

Comments to the Author:

As you will see, both reviewers found merit in your systematic approach to replicating previous work but they also both criticise your interpretation of findings (other people's and yours). This manuscript is not the place to discuss in detail all relevant literature in this debate but I urge you to provide a more balanced review of the literature as well as the additional data and clarifications requested by the reviewers especially with respect to task modifications you have made that could have affected performance. Also please make sure to also provide explanations for the failures of standard tasks.

Reviewers' Comments to Author:

Reviewer: 1

Comments to the Author(s)

This manuscript reports three studies, attempting to replicate and extend an interactive paradigm by Southgate et al. (2010) that was meant to tap into infants' and toddlers' theory of mind abilities. Results using a close approximation of the original study failed to replicate its main finding (that 17-month-olds seemed to understand an actor's false belief), and the additional studies with various manipulations to attempt to make the task easier/more pragmatically felicitous also failed to demonstrate that toddlers (and even 3- and 4-year-olds) would respond in accord with an understanding of false belief. Adults, however, responded as expected on the "Sefo" task and their performance was correlated with other theory of mind measures.

This is a well-written paper, reporting a programmatic series of studies. I appreciated that the authors went beyond demonstrating a "failure to replicate" by attempting to modify the task still further and to investigate e.g., its relation to other tasks. I have a few comments and suggestions.

1. My primary comment is that I believe the authors have demonstrated conclusively that they were unable to replicate findings from the original task. But this leaves open the question (which I don't think the authors address, though I may have missed it) about why the discrepancy? Is it because of the small sample size in the original study? Were the authors able to confirm with the original researchers that the procedure was done in exactly the same way? As I know the authors are aware, failures to replicate can occur for many reasons and it would be good to be able to work with the original researchers to try to understand together why their findings were different.
2. I would also encourage the authors to consider further the implications of their findings. They argue persuasively that whether toddlers understand false beliefs has important theoretical implications re nativism, for example. This seems reasonable. But it seems like overstating the case to suggest that theory of mind reduces to an understanding of false beliefs, which I don't think the authors want to argue but which is an interpretation a reader could make. Additionally, I'd encourage the authors to try to link the abilities they believe toddlers have or do not have to children's real-world experience. What would it mean (beyond performance on experimental tasks or theoretical debates about nativism) for children to have the kind of understanding that some scientists attribute to them?
3. I wondered whether the authors could code from video other aspects of the children's

performance that could also speak to some of the debates in the field. For example, were children slower to make responses in the TB than FB conditions (even though they ended up being at chance in which object they chose)? Did they look more at one than the other object/box depending on condition? What kind of spontaneous comments (if any) did children make – did they request clarification? What kinds of individual differences might obtain in these kinds of tasks?

Reviewer: 2

Comments to the Author(s)

Review of RSOS-191998

Entitled 'Actions do not speak louder than words in an interactive false belief task'

Overview:

This paper presents 3 experiments with the expressed objective to replicate (closely and with modifications) the study of Southgate, Chevallier and Csibra, 2010. The paper mainly focuses on the relevance of the replication approach and on the difficulties in finding robust and reliable methods for studying the development of theory of mind. No doubt, replicability is a central issue for research lead by the idea of accumulating valid knowledge, and psychology is an area of research facing a serious replication crisis. That said, it is extremely important to pursue transparent and stringent research practices in such replication attempts as well and at the same time the theoretical framing should remain objective.

I appreciate this general approach together with the efforts the authors invested into empirical testing, nonetheless, I have concerns with respect to their presentation of data and the conclusions that can be drawn from them. I miss important details in the presented methods and results and find the interpretation of the selected pattern of data somewhat biased.

In order to this paper become more stringent, persuasive and constructive, presentation of additional methodological details, data and a more toned down interpretation is proposed.

Concerns in details:

Introduction:

My main impression is that the authors handle replication problems and theory driven debates on the interpretation of given findings as the two facets of the same problem (the authors handle these as reliability and validity of the empirical phenomenon). While both questions have the highest importance, and clearly they are related, these issues cannot be directly equated. The entire introduction suffers from this latter issue (the validity of a phenomenon is not always handled correctly in my opinion). Bringing up replication attempts and their results as the main focus of the paper seems to justify the selection to include only those studies that has been targeted to be replicated. Clearly, this approach results in a selective set of references. This is fine unless someone does want to question the underlying theoretical constructs as well, in which case the related conceptual approaches (though not necessarily being replications) would be important to be mentioned and considered in details. Otherwise the reader receives a biased picture of the state of the art research in the field. So, if the authors' objectives entail the discussion of theoretical approaches as well, they ought to include the contrastive explanatory angle and the empirical bases of them too.

Let me highlight this problem with the help of some examples:

1. One of the the authors' main claims is:

'In fact, a growing body of replication studies puts into question the robustness of the original evidence on early and automatic ToM (for an overview, see Kulke & Rakoczy, 2018).' p. 5

Following this claim, the authors mention the failed replications of both anticipatory looking tasks and looking time studies, however they ignore the commentary that focuses on the problems with the above replication attempts, see Baillargeon, Buttelmann, Southgate, 2018. Also,

the authors do not mention that a collection of conceptual replications are reviewed in the literature (see Scott and Baillargeon, 2017). The purpose of representing the debate itself - and not only their own take on - would necessitate discussing the other viewpoint, or at least referring to it.

2. Here is another example on what kind of phrases and claims I find misleading: ‘Similarly, ToM-based priming effects in implicit tasks with adults could recently also not be replicated (Conway, Lee, Ojaghi, Catmur & Bird, 2017).’ p. 5.

This sentence suggests that implicit ToM-based priming tasks in general are not replicable. There are several problems with this claim then. First, a closer look at the referred paper reveals that the objective of the study is to offer a competing interpretation for an empirically reliable (replicable) phenomenon. So, while providing a conceptual replication of the phenomenon, the paper aims to disentangle different explanations beyond it. That is true, the findings seem to support a non-mentalistic interpretation of the data, yet this study is NOT an exemplar of a non-replication. Second, even if the role of the study in the paper was to represent problems with replicating similar studies, the authors should not dismiss that the actual phenomenon has been replicated by couple of other studies conceptually, with contrasting interpretation to the one mentioned (see Furlanetto et al., 2016; Marshall et al. 2018; Wiese et al., 2012; Teufel et al., 2010), and there are other paradigms on perspective taking with similar theoretical background that are actually highly replicable across laboratories (Surtees et al., 2016; Elekes et al., 2016.)

3. With respect to interactive FB tasks, the manuscript focuses on the so called helping task by Buttelmann et al., 2009 and the attempts to replicate it. The argument in the manuscript is that while this empirical paradigm is at least partially replicable, there is a problem with the validity of the original task, as it ‘may not measure belief-tracking but simpler forms of social cognition such as goal-understanding.’ p. 6

I am fine with the claim that the responses given in the experimental protocol can be interpreted differently, however the dispute would be mentioned in more details, to make obvious for the readers in what sense the validity of the protocol is in question. I am emphasizing this as the alternative interpretation itself also received criticism, namely that the teleological approach mentioned can explain how children bring up reasons for an already performed behavior (not necessarily ‘a simpler form of social cognition’), but based on that approach there is no answer on how and whether children are able to predict the behavior of a protagonist -the original problem of ToM research (see <http://cognitionandculture.net/blogs/pierre-jacob/if-teleology-is-the-answer-what-was-the-question/>).

4. Arriving to the focal purpose of the paper, the authors introduce the sefo task. I miss here the detailed description of the Loth and Happé task (2002) and also of the task of Carpenter et al (2002). Though done with 3 year olds, those studies served as sources for the sefo task and actually provide convergent evidence that 3 year-olds are able to exploit belief ascription for the sake of word learning.

5. In relation to the sefo task the already existing replication attempts are discussed. Here, I would like to suggest a careful and thorough description and interpretation of data in general, as this is the most helpful strategy to support open science practices and the understanding of the robustness of the phenomenon.

So, with respect to the Grosse Wiesmann et al. (2017) study, it would be important to mention that the main question in such a task is whether the response pattern in the FB condition shifts away from children’s natural (baseline) response to a request accompanied by a pointing gesture which is measured in the TB condition. Therefore, a comparison between TB and FB performance is needed in order to grasp the modulatory effect of belief ascription in the interaction, that is actually missing in the above experiment (while that is true that the original study could also show above-chance performance for the FB conditions).

With respect to the Kiraly et al (2018) study the authors raise: 'a main concern is that this finding is difficult to interpret given that no analogous prospective TB condition was conducted.' I do not see the main concern here. Actually, there was a TB condition that was used as a bases for comparison for the prospective FB condition as well: at the beginning of the protocol infants explored a pair of sunglasses that were see through, and then they saw the model using an other pair of sunglasses. True, that in this condition children got themselves information that those pair of sunglasses the model wore were transparent as well. Is there any theory that would predict difference in performance just based on receiving confirmatory information later? In what sense a 'just prospective TB condition would modify the results? So, this concern is of no base, it is simply misleading.

Clearly true that this conceptual replication contained modifications that were difficult for young children in the specific TB-FB condition in terms of memory load, where 18-mo-olds failed, but that was the target question of the paper of Kiraly et al. (children had to keep track of the status of the sunglasses and relate that to the possible belief content of the protagonist with the help of their memory capacities). What is the relevance then of the description of this modification in the present paper without mentioning the target question of the original paper? Again, this is somewhat misleading, as it is presented as a concern.

In addition, both for the Grosse Wiesmann study and the Kiraly et al. study the argument is brought up that the absence of the deceptive motive could have led to the low performance in the FB condition. Is it the claim that just based on the fact that the model remained in the room during the test in both studies the deceptive mode is questionable? In my opinion, as actually the deceptive mode was performed in both studies while the model was present, the authors should also try to give an explanation on how children interpret the deceptive mode in the presence of the model. So, the suggestion is either give more details on this explanation or leave out the post hoc interpretation, as it cannot give a systematic explanation (same context resulted in failure in 3- and 4-year-olds in Grosse Wiesmann et al study, failure in 18-mo-olds in Kiraly et al study, yet in success in 3-year-olds in Kiraly et al study).

The concerns with respect to the linguistic complexity used for older children are also somewhat confusing. If I understand correctly, the argument is that in case of the study of Kiraly et al., the use of the original phrases from Experiment 1 of Southgate et al. (Do you remember what I put here?) would have induced a pragmatic shortcut for 3-year-olds to solve the task without belief ascription. But only for the FB condition, since, according to the authors: 'due to different pragmatic contexts between conditions, children may interpret the prompt in the TB condition differently'. What are those differences in the pragmatic context (for the TB and FB condition) that are completely unrelated from belief ascription? Please describe the details, I do not find them obvious. Rather, this claim would predict pragmatic ambiguity for the TB condition as well if children do not take into account the belief of the speaker. (Note again, the original study did show the same pattern of results with three different prompting).

Study 1: Direct replication

Methodological issues

1. I find compelling that '40% of the 17-months-olds tested in Study 1 struggled to pass the original familiarization criterion'. p 16. The authors therefore decided to a) modify the procedure slightly for some participants, and b) applied a more lenient passing criteria for a subsample of participants. One can find a sample description in the result section, referring to the lenient criteria passers: FB: n = 12, TB: n = 8. So then, the sample of children who passed the original criterion (FB: n = 12, TB: n = 16) includes those participants for whom the procedure was modified slightly or not? This particular detail could be of importance to evaluate the discussion of the difficulties of familiarization in relation to the reliability of the phenomenon. The authors argue for the reliability of the protocol by emphasizing that there was no performance difference between the subsamples that did vs. did not reach the familiarization criterion. This claim would receive more emphasis if the subsample of passers would not include those who received modified demonstrations.

(The claim that for 3-year-olds the exact same familiarization did not provide any challenge while they also failed in the test is inconclusive already because of the potential age differences: the fact that there was no problem for older children does not rule out that there could have been issues for younger ones. Nonetheless, actually there was a tendency difference in the performance in TB vs FB conditions in the older age-group).

2. There are some missing minor methodological details by the clarification of which the methods would be more clear. When did E2 introduced the deceptive mode? Already at the beginning of the location swap episode? Was this deception mode there for the TB condition as well? How long was E1 away? How many pointing were shown in the test phase (just one or more)?

3. While the authors explicitly declare that Study 1 is a direct and as close as possible replication of the original study they modify the verbal prompts used in the task. The verbal prompts of Experiment 2 and Experiment 3 of the original study were combined. The result is: "Do you know what's in here? I want to play with this"... "Can you give it to me?". (Just for comparison, verbal prompts of the original study: Exp 2 of Southgate et al.: Do you remember what I put in here? Shall we play with it? Shall we play with it? Let's play with it! Finally, she said, 'Can you get it for me?' Exp 3: Do you know what's in here? There's a sefo in the box! There's a sefo in here! Can you get it for me?)

The authors themselves call attention to the potential effect of the pragmatic ambiguity of some verbal prompts that can overshadow children's belief ascription capabilities, (likewise they argue for the case of the prompt used by Grosse Wiessman et al, 2017; see page 8.) I myself think that while the purpose of this new prompt could have been to use the most neutral phrasing, the result of the choice is again equivocality. The assumption beyond the interactive FB task is that word label mapping or reference mapping trigger belief ascription: in search for a correct mapping of the referent of the interlocutor's request the child need to track the belief state of her. However, what is the message of the prompts used in this study? The model asks whether the CHILD knows what is in the box, whatever it is, and clearly, she expresses she would like to play with that object. In contrast, the versions of the original study highlight the possible mismatch between the protagonist's perspective and the child's knowledge, either by referring to a previous event and a previous state of the scenario (using the phrase 'Do you remember what I put in here?') or by introducing a novel label, for the mapping of which the child needs to track the protagonist mental states (she cannot know that the object in here is the SEFO). So, the pragmatic call of the phrase in the present version is different from that of the originals. I think it is highly relevant to acknowledge this difference in the paper. Just saying that this prompt is similar to the original version of Exp 3 is inaccurate. Based on this, it is questionable to call the study as direct replication.

4. The study introduced three consecutive trials for testing 17-mo-olds where feedback was given. The text describes the forms of feedback for the FB conditions, but what was the feedback for the TB conditions?

5. It is unclear from the text that in the case of the 3-year-old group each child was tested in only TB conditions or FB conditions? That is what the following sentence means? „3 -year-olds were tested with a single Sefo FB/TB test trial and one trial of a corresponding standard FB/TB (change-of-location) task (task order counterbalanced).”

Results

1. What is the exact reason for collapsing the sample and check whether there is a tendency to choose the referred box?

2. Can the authors report a matrix of failers/passers with respect to the sefo and the standard task? If I understand correctly, this data is available, though only separately for TB and FB conditions.

3. I doubt that a correlation analysis is well grounded when the data for the to-be-correlated measures on both sides show random pattern!! At least, no correlation in such case is definitely inconclusive.

Discussion

1. Is there any explanation on why there was no learning effect in the TB condition for 17-month-olds?
2. What does the standard TB version measure?
3. In discussion of the performance of the 3-year-olds, (who actually performed at chance in both standard FB and TB tasks), instead of accepting that children were simply guessing in both kinds of tasks, the interpretation is raised that there are two separate subsamples (FB passers/TB non-passers and TB passers/FB non-passers in the group of 3-year-olds. This interpretation sounds very speculative. Is there any solution to prove it based on the data? Otherwise, based on Occam's razor, (and also based on the data on TB non-passers introduced in the paper they cite) it is more likely that the study represents a no replication of the TB version of the classic task.

Study 2a

Methodological issues

1. Note, the pragmatic ambiguities revealed by the verbal prompts hold fold the 'original' sefo task here as well, as described above.
2. Actually, in the modified task the verbal prompting is different, it contains a new verbal labelling for one of the new objects as in Exp3 of Southgate et al. However, based on the description, the experimenter continuously pointed at the box during the verbal prompting. This is again a difference from the original study. Was there a difference in this respect between Study 1 and the 'original' trials of Study 2a? Worth to mention this detail, as this strong referential cue can overwrite children's knowledge based inferences (as the authors themselves discuss this possibility in the general discussion, but not as part of giving the rationale for Study 2b).
3. Were there any feedback or related verbal description between the trials?

Results

Clearly, in all conditions children followed the pointing behaviour of the experimenter and chose the object from the referred box. I think that there is no reason to calculate correlation in case of such homogenous pattern at ceiling.

Study 2b

I find the objective and the implementation of this task very interesting. Ecologically it seems to be more valid, the request of the experimenter emerges from an obvious need. However, I still have doubts whether this implementation is pragmatically more transparent than the previous ones. In my reading, the familiarization of the task assures children that the experimenter knows what makes an apparatus work. When the experimenter is back and deliberately referring to one of two boxes -each of which contains a novel object -, and says 'We need what is in there. Can you give it to me?', this can be taken as an assertion that they (the child and the experimenter) need the specific object that is in the box the experimenter actually is referring to. As previously she turned out to be knowledgeable, no doubt emerges in the situation.

I would like to underline again that in previous versions a tension between the referential cuing and the assertion was present ('remember' or the need to map a novel label to an object).

Study 3

The question remains open in what context infants literally follow the strong referential cuing and when they are sensitive to the belief of the model. Study 3 proposes to investigate this

question. The general introductory argument is that there was a repetitive failure in finding evidence that the sefo task is measuring whether young children keep track of other agents' beliefs in communicative interactions. The theoretical perspective can be formed differently: introducing modifications to the sefo task may alter the interpretation of the ongoing interaction, and consequently, the interpretation of the communicative intention of the partner by infants: in most of the cases they are led by the teaching intent of the protagonist, when it occurs within highly ostensive cues. The discussion and introduction would benefit from raising this alternative (Csibra and Gergely, 2009).

There is no clear description on what the explicit question versions of the Sefo task intended to measure. Indeed, they offer a possibility to answer the question whether children reacted differently in the behavioural request situation and in the label and belief questions. As those answers are coming from the same participants, a related sample analysis is suggested: that could show that there were differences in the answer patterns for the different tasks, thus potentially different interpretation of the request and the questions on the part of the same participants. I think the McNemar test fails to grasp the change between choices within participants.

Still, there is no discussion on the fact that the same children intended to follow directly the referential request of the experimenter, while their belief mapping and labelling attempts were significantly different from their behavioural choice. What is the explanation for this?

General discussion

I completely agree with the authors that the Sefo task in the future need to be more systematically investigated and tested in multilab collaboration. However, before arriving to this conclusion, it would be worth to summarize that even in the present paper there were some inconsistencies in result patterns (chance level performance for 17-mo olds for both FB and TB sefo conditions in Study 1, tendency difference for 3 year olds between FB and TB conditions in Study 1; always choosing the referred box irrespective of condition at ceiling in Study 2; difference in the pattern of responses for behavioural request and the verbal questions in the sefo trials in Study 3) that highlights that the sefo task as being an interactive task, the focus of interaction can be shifted sensitively to contextual factors.

The problems with the standard TB task should also be mentioned in the general discussion.

The discussion of the factor that the pattern of data in Study 2 and Study 3a is similar to the tendency found in the literature that 3- to 4-year-olds exhibit a robust bias to search in pointed-to locations even when the pointer has turned out to be unreliable, deceptive or ignorant about the object location deepens the need to study further what influence the behavior of children in such interactive tasks.

References:

Baillargeon, R., Buttelmann, D., Southgate, V. (2018). Invited Commentary: Interpreting failed replications of early falsebelief findings: Methodological and theoretical considerations. *Cognitive Development* 46. 112-124

Csibra, G., Gergely, Gy. (2009) *Natural Pedagogy*. *Trends in Cognitive Sciences* 13(4):148-53

Elekes, F., Varga, M., & Király, I. (2016). Evidence for spontaneous level-2 perspective taking in adults. *Consciousness and Cognition*, 41, 93-103.

Furlanetto T, Becchio C, Samson D, Apperly I. (2016). Altercentric interference in level 1 visual perspective taking reflects the ascription of mental states, not submentalizing. *J Exp Psychol Hum Percept Perform*. 42(2):158-63.

Marshall, J., Gollwitzer, A., Santos, L.R. (2018). Does altercentric interference rely on mentalizing?:

Results from two level-1 perspective-taking tasks. *PLoS One*, 13(2), e0194101.
doi: 10.1371/journal.pone.0194101

Scott, R. M., & Baillargeon, R. (2017). Early false-belief understanding. *Trends in Cognitive Sciences*, 21(4), 237-249.

Surtees, A., Apperly, I., & Samson, D. (2016). I've got your number: Spontaneous perspective-taking in an interactive task. *Cognition*, 150, 43-52.

Teufel, C., Alexis, D. M., Clayton, N. S., & Davis, G. (2010). Mental-state attribution drives rapid, reflexive gaze following. *Attention, Perception & Psychophysics*, 72(3), 695-705.

Wiese, E., Wykowska, A., Zwickel, J., & Müller, H. J. (2012). I see what you mean: How attentional selection is shaped by ascribing intentions to others. *PLoS One*, 7(9), e45391.

Author's Response to Decision Letter for (RSOS-191998.R0)

See Appendix A.

RSOS-191998.R1 (Revision)

Review form: Reviewer 1

Is the manuscript scientifically sound in its present form?

Yes

Are the interpretations and conclusions justified by the results?

Yes

Is the language acceptable?

Yes

Do you have any ethical concerns with this paper?

No

Have you any concerns about statistical analyses in this paper?

No

Recommendation?

Accept with minor revision (please list in comments)

Comments to the Author(s)

This revision reports an attempted replication and extension of research on false belief understanding in toddlers. I think the revised version has addressed several issues and I appreciate that the authors have explicitly offered suggestions for why their findings differ from those of the original study. I offer a few additional comments below.

1. I continue to have concerns that the authors are conflating "theory of mind" with an understanding of false belief. For example, on p. 4 of the revision, they write, "a growing body of

replication studies puts into question the robustness of the original evidence on early ToM." But as noted in my review of the original manuscript, it seems to me that what the authors have concerns about is specifically the "robustness of the original evidence on early FB". I recognize that the authors believe that FB is a "litmus test" for ToM. But I'm not sure how widely held this perspective is: It seems to me that most folks seem to believe that ToM involves a suite of social cognitive skills; an understanding of FB is but one (a point the authors seem to make on p. 5 when they attribute findings from Buttelmann et al., 2009, to "goal-understanding"). One solution would be to make clear it what the authors are objecting to and arguing by e.g., changing "ToM" to "FB" throughout (which would also match the title, which is specific to false belief).

2. The explanation of the Sefo task was unclear in the distinction between Study 1 and Study 2. I think for Exp. 2, it would be helpful to provide the entire language used to make it clear what was the same and what was different from Study 1. That is, I'm inferring that in Study 2, the E said, "Do you remember what I put in here? Shall we play with it? Can you get it for me?" But the "Do you remember what I put in here" is not included in the text for Study 2.

3. The order of events in the procedure described for Study 2 on p. 21 could be clearer. That is, the information about what happens after the swap in terms of how E1 asks for objects is provided before the information about what happens before the swap in terms of how E1 puts the objects in the boxes and E2 changes their positions.

Review form: Reviewer 3

Is the manuscript scientifically sound in its present form?

Yes

Are the interpretations and conclusions justified by the results?

No

Is the language acceptable?

Yes

Do you have any ethical concerns with this paper?

No

Have you any concerns about statistical analyses in this paper?

No

Recommendation?

Reject

Comments to the Author(s)

'Actions do not speak louder than words in an interactive false belief task'
Royal Society Open Science

This manuscript reports three experiments trying to replicate (Study 1), extend (Study 2) and validate (Study 3) the results originally observed by Southgate and colleagues using the so-called 'Sefo task' (2010). While I did not review the first submission, I could see from the authors' response letter that R1 and R2 had raised important points and offered detailed and careful comments. In my review, I will address a major concern I have about Study 1, point out a couple of smaller points, and close with a general comment about failed replications.

[1]

In Study 1, which was supposed to be a direct replication of the original study, the authors had problems getting infants to pass the warm-up trials, to the point of having to relax the inclusion criteria used by Southgate et al., and in some cases leave open the boxes for the infants to reach for the objects (see p. 10). When discussing the results of their first study, the authors acknowledge this shortcoming (see pp. 17-18). However, they rule out the possibility that this issue could explain their null results on the following grounds:

“We did not find performance differences between the subsamples that did vs. did not reach the familiarization criterion. Furthermore, all three-year-old children passed the original familiarization criterion and still performed at chance level in the FB condition. Therefore, it seems unlikely that the low performance in the familiarization trials led to the low performance in the test scenario and thus the failed replication of the original results.”

While the above arguments are sound, this discussion fails to acknowledge a more worrying possibility: infants’ poor performance in the warm-up trials could reveal a lack of engagement in the task, or even a lack of rapport with the experimenter (something that would be key to pass this task, since participants need to understand the experimenter’s goal). If this problem was general enough, infants could have underperformed across the board, which would explain their chance performance (note that the infants in Study 1 did not show a preference for the referenced box, unlike the other age groups). The above statistical comparisons, which the authors use to defend the validity of their findings, do not address such a problem.

I suggest that the authors acknowledge that infants’ poor performance in the warm-up trials (something that was not observed by Southgate et al. (2010), nor in Studies 2 and 3 in this paper) could be symptomatic of problems in engaging the infants in the task, which could explain their chance performance without casting doubts on the validity of Southgate et al.’s results.

[2]

In addition, Study 1 is reported as a ‘direct replication’ of Southgate et al. (2010). However, as the authors acknowledge on p.11, they mixed different features of the original experiments, which in my view doesn’t qualify as a direct replication. I appreciate that Study 2 departed more radically from the original design, but I still think it’s confusing to refer to Study 1 as a direct replication when it did not use the protocol of any of the original experiments.

[3]

More generally, I think the authors should acknowledge that they have had problems getting their participants to meet the original inclusion criteria of the studies they have tried to replicate. The low performance observed in warm-up/familiarization trials could be indicative of an overall lower reliability of the critical trials – a possibility that is not refuted by comparing the performance of children who pass and fail the inclusion criteria.

Baillargeon, Buttelmann and Southgate already addressed this issue in their 2018 commentary in *Developmental Psychology*:

"Table 1 reveals clear differences between the rate at which participants (infants, older children, and adults) met this inclusion criterion between the different versions of the paradigm. For example, whereas 64% of 25-month-olds exhibiting an anticipatory saccade in Southgate et al. (2007) correctly predicted the location of the agent’s search by the second familiarization trial, only 25% and 34% of infants did this in Schuwerk et al. (2018) and Grosse Wiesmann et al. (2018), respectively. Similarly, Senju et al. (2009) and Low and Watts (2013) both reported that 100% of their adult participants (and for Low & Watts, also 3- and 4-year-olds) showed correct prediction by the last familiarization trial, whereas only 45% of adult participants

reached this criterion in the Kulke et al. (2018b) study. These low inclusion rates meant that in order to reproduce the original analyses, these papers had to make inferences based on samples that were of a comparably small size to the original study (and in one case from 9 infants, Kulke et al., 2018a).

To what extent these differences have implications for the interpretation of test trial failures is unknown. When all (Low & Watts, 2013; Senju et al., 2009) or most (Senju et al., 2010; Senju et al., 2011; Southgate et al., 2007) participants pass the second familiarization trial, one could conclude that as a group, they understood the task and were motivated to make a prediction. When only half the sample exhibit a correct anticipatory look on the second familiarization trial, it is legitimate to ask whether what looks like success in these 50% is actually success, or rather just random looking. The lights in the window will draw participants' attention to one of the windows so there is a 50% chance of being 'correct' without anticipating action. Thus, it is not straightforward to use equivalent test trial failure in those who passed the second familiarization trial and those who did not as justification for abandoning the original inclusion criteria (Dörrenberg et al., 2018). To compare, in their meta-analysis of the traditional false-belief task, Wellman et al. (2001) excluded from their analysis any studies in which fewer than 60% of children answered the comprehension- check control questions correctly."

[4]

I understand from the reviewers' comments that the original manuscript came across as biased. While I think the authors have done a good job in revising the manuscript, the abstract continues to be one-sided: the authors only refer to failed replications of the Sefo task, failing to acknowledge that the results with that task have been mixed.

[5]

The literature review includes an inaccuracy on p. 4:

"These findings on infants' FB understanding and adults' automatic tracking of others' perspectives had far-reaching implications and laid the grounds for novel theoretical approaches to ToM development, including nativism (Scott & Baillargeon, 2017) and two-system-accounts (Apperly & Butterfill, 2009)."

Nativist accounts of Theory of Mind have been around since the birth of the field in the 80's. In fact, Onishi and Baillargeon (2005) motivated the first successful study with infants as an investigation of those early nativist accounts (citing Alan Leslie and Jerry Fodor, amongst others).

While I appreciate that the authors want to emphasize the implications of false-belief studies with infants for the field, they should not overstate them.

[6]

Finally, I would like to raise the question of what exactly this study contributes to the literature. As the authors now acknowledge in the introduction, three recent studies have already shown that the results of the Sefo task have been mixed (Grosse Wiesmann et al., 2017; Dörrenberg et al., 2018; Kiraly et al., 2018). Moreover, the authors conclude the paper with the following desiderata for future research:

"In future research, the reliability and validity of the Sefo and related interactive ToM tasks need to be investigated in systematic and in-depth ways, ideally in a multi-lab, preregistered collaborative endeavor such as ManyBabies (Frank et al., 2017)."

Given that we already know from previous studies that the results of the Sefo task have been mixed, and the ManyBabies project is already well underway, why continue to flood the market

with failed replications that do not provide any conclusive evidence as to why the results of the original studies have not always replicated?

For what is worth, I want to copy here my review of another paper that this group published recently with Royal Society Open Science (led by Louise Kulke, who has made an entire career out of publishing failed replications – and failed replications alone), as I think it shows how the ‘publication bias’ that the authors seem so concerned about (see p. 38) is in fact a signature of their own work in recent years:

“Related to the last point, we should consider what would have happened if this study had been an original study, and not a conceptual replication of previous work. In view of their null results, the authors would have had to admit – as most of us have had to admit often enough – that their study had failed: the manipulations they had carefully introduced to tap a certain effect simply didn't work. However, while researchers addressing new questions with new paradigms face the risk of failing, researchers only aiming to replicate previous studies seem to have found a new business model in Academia: if after a few failed replications, researchers finally manage to replicate the original results, the replicated results would be news and therefore publishable material. However, if their manipulations failed (as they clearly did here), they can always write up a new failed replication and continue to question previous findings, while not adding anything new to the literature.

I think this practice is highly questionable as it makes failed replications immune to failure (ironically) and always publishable, regardless of the possible shortcomings that would prevent publication of original studies. Since publication standards should be just as high for original work and replication studies, I cannot recommend this manuscript for publication.”

Decision letter (RSOS-191998.R1)

Dear Mrs Wenzel:

I write you in regards to manuscript # RSOS-191998.R1 entitled "Actions do not speak louder than words in an interactive false belief task" which you submitted to Royal Society Open Science.

Regrettably, in view of the criticisms of the reviewer(s) found at the bottom of this letter, your manuscript has been denied publication in Royal Society Open Science.

Thank you for considering Royal Society Open Science for the publication of your research. I hope the outcome of this specific submission will not discourage you from the submission of future manuscripts.

Royal Society Open Science
openscience@royalsociety.org
on behalf of Dr Teodora Gliga (Associate Editor) and Essi Viding (Subject Editor)
openscience@royalsociety.org

Associate Editor Comments to Author (Dr Teodora Gliga):

Comments to the Author:

I have now received comments from two reviewers, one that had seen the manuscript before and an additional reviewer that has only seen your re-submission. While the first reviewer recommends publication, they continue to question what this study adds to our understanding of cognitive development. The second reviewer is more critical, mainly questioning what this study adds to a body of work that is already very mixed.

Views in this field seem very polarised and I agree with the reviewers that your manuscript is limited in terms of advancing our understanding. I therefore decide to reject this paper on the following grounds: 1. because it only questions the ability of a particular task to capture belief understanding and not the presence of this skill, therefore having limited scope in terms of advancing our understanding of cognitive development and 2. because you do not provide additional information for why this task yields mixed positive and negative replications thus adding little in terms of understanding the conditions under which young children may or may not manifest an understanding of false beliefs.

I know this decision is disappointing especially given all the work you have put into addressing reviewer's comments and I do hope you will consider RSOS for future submissions.

Reviewer comments to Author:

Reviewer: 1

Comments to the Author(s)

This revision reports an attempted replication and extension of research on false belief understanding in toddlers. I think the revised version has addressed several issues and I appreciate that the authors have explicitly offered suggestions for why their findings differ from those of the original study. I offer a few additional comments below.

1. I continue to have concerns that the authors are conflating “theory of mind” with an understanding of false belief. For example, on p. 4 of the revision, they write, “a growing body of replication studies puts into question the robustness of the original evidence on early ToM.” But as noted in my review of the original manuscript, it seems to me that what the authors have concerns about is specifically the “robustness of the original evidence on early FB”. I recognize that the authors believe that FB is a “litmus test” for ToM. But I’m not sure how widely held this perspective is: It seems to me that most folks seem to believe that ToM involves a suite of social cognitive skills; an understanding of FB is but one (a point the authors seem to make on p. 5 when they attribute findings from Buttelmann et al., 2009, to “goal-understanding”). One solution would be to make clear it what the authors are objecting to and arguing by e.g., changing “ToM” to “FB” throughout (which would also match the title, which is specific to false belief).
2. The explanation of the Sefo task was unclear in the distinction between Study 1 and Study 2. I think for Exp. 2, it would be helpful to provide the entire language used to make it clear what was the same and what was different from Study 1. That is, I’m inferring that in Study 2, the E said, “Do you remember what I put in here? Shall we play with it? Can you get it for me?” But the “Do you remember what I put in here” is not included in the text for Study 2.
3. The order of events in the procedure described for Study 2 on p. 21 could be clearer. That is, the information about what happens after the swap in terms of how E1 asks for objects is provided before the information about what happens before the swap in terms of how E1 puts the objects in the boxes and E2 changes their positions.

Reviewer: 3

Comments to the Author(s)

'Actions do not speak louder than words in an interactive false belief task'

This manuscript reports three experiments trying to replicate (Study 1), extend (Study 2) and validate (Study 3) the results originally observed by Southgate and colleagues using the so-called 'Sefo task' (2010). While I did not review the first submission, I could see from the authors' response letter that R1 and R2 had raised important points and offered detailed and careful comments. In my review, I will address a major concern I have about Study 1, point out a couple of smaller points, and close with a general comment about failed replications.

[1]

In Study 1, which was supposed to be a direct replication of the original study, the authors had problems getting infants to pass the warm-up trials, to the point of having to relax the inclusion criteria used by Southgate et al., and in some cases leave open the boxes for the infants to reach for the objects (see p. 10). When discussing the results of their first study, the authors acknowledge this shortcoming (see pp. 17-18). However, they rule out the possibility that this issue could explain their null results on the following grounds:

"We did not find performance differences between the subsamples that did vs. did not reach the familiarization criterion. Furthermore, all three-year-old children passed the original familiarization criterion and still performed at chance level in the FB condition. Therefore, it seems unlikely that the low performance in the familiarization trials led to the low performance in the test scenario and thus the failed replication of the original results."

While the above arguments are sound, this discussion fails to acknowledge a more worrying possibility: infants' poor performance in the warm-up trials could reveal a lack of engagement in the task, or even a lack of rapport with the experimenter (something that would be key to pass this task, since participants need to understand the experimenter's goal). If this problem was general enough, infants could have underperformed across the board, which would explain their chance performance (note that the infants in Study 1 did not show a preference for the referenced box, unlike the other age groups). The above statistical comparisons, which the authors use to defend the validity of their findings, do not address such a problem.

I suggest that the authors acknowledge that infants' poor performance in the warm-up trials (something that was not observed by Southgate et al. (2010), nor in Studies 2 and 3 in this paper) could be symptomatic of problems in engaging the infants in the task, which could explain their chance performance without casting doubts on the validity of Southgate et al.'s results.

[2]

In addition, Study 1 is reported as a 'direct replication' of Southgate et al. (2010). However, as the authors acknowledge on p.11, they mixed different features of the original experiments, which in my view doesn't qualify as a direct replication. I appreciate that Study 2 departed more radically from the original design, but I still think it's confusing to refer to Study 1 as a direct replication when it did not use the protocol of any of the original experiments.

[3]

More generally, I think the authors should acknowledge that they have had problems getting their participants to meet the original inclusion criteria of the studies they have tried to replicate. The low performance observed in warm-up/familiarization trials could be indicative of an

overall lower reliability of the critical trials – a possibility that is not refuted by comparing the performance of children who pass and fail the inclusion criteria.

Baillargeon, Buttelmann and Southgate already addressed this issue in their 2018 commentary in *Developmental Psychology*:

"Table 1 reveals clear differences between the rate at which participants (infants, older children, and adults) met this inclusion criterion between the different versions of the paradigm. For example, whereas 64% of 25-month-olds exhibiting an anticipatory saccade in Southgate et al. (2007) correctly predicted the location of the agent's search by the second familiarization trial, only 25% and 34% of infants did this in Schuwerk et al. (2018) and Grosse Wiesmann et al. (2018), (personal communication), respectively. Similarly, Senju et al. (2009) and Low and Watts (2013) both reported that 100% of their adult participants (and for Low & Watts, also 3- and 4-year-olds) showed correct prediction by the last familiarization trial, whereas only 45% of adult participants reached this criterion in the Kulke et al. (2018b) study. These low inclusion rates meant that in order to reproduce the original analyses, these papers had to make inferences based on samples that were of a comparably small size to the original study (and in one case from 9 infants, Kulke et al., 2018a).

To what extent these differences have implications for the interpretation of test trial failures is unknown. When all (Low & Watts, 2013; Senju et al., 2009) or most (Senju et al., 2010; Senju et al., 2011; Southgate et al., 2007) participants pass the second familiarization trial, one could conclude that as a group, they understood the task and were motivated to make a prediction. When only half the sample exhibit a correct anticipatory look on the second familiarization trial, it is legitimate to ask whether what looks like success in these 50% is actually success, or rather just random looking. The lights in the window will draw participants' attention to one of the windows so there is a 50% chance of being 'correct' without anticipating action. Thus, it is not straightforward to use equivalent test trial failure in those who passed the second familiarization trial and those who did not as justification for abandoning the original inclusion criteria (Dörrenberg et al., 2018). To compare, in their meta-analysis of the traditional false-belief task, Wellman et al. (2001) excluded from their analysis any studies in which fewer than 60% of children answered the comprehension-check control questions correctly."

[4]

I understand from the reviewers' comments that the original manuscript came across as biased. While I think the authors have done a good job in revising the manuscript, the abstract continues to be one-sided: the authors only refer to failed replications of the Sefo task, failing to acknowledge that the results with that task have been mixed.

[5]

The literature review includes an inaccuracy on p. 4:

"These findings on infants' FB understanding and adults' automatic tracking of others' perspectives had far-reaching implications and laid the grounds for novel theoretical approaches to ToM development, including nativism (Scott & Baillargeon, 2017) and two-system-accounts (Apperly & Butterfill, 2009)."

Nativist accounts of Theory of Mind have been around since the birth of the field in the 80's. In fact, Onishi and Baillargeon (2005) motivated the first successful study with infants as an investigation of those early nativist accounts (citing Alan Leslie and Jerry Fodor, amongst others).

While I appreciate that the authors want to emphasize the implications of false-belief studies with infants for the field, they should not overstate them.

[6]

Finally, I would like to raise the question of what exactly this study contributes to the literature. As the authors now acknowledge in the introduction, three recent studies have already shown that the results of the Sefo task have been mixed (Grosse Wiesmann et al., 2017; Dörrenberg et al., 2018; Kiraly et al., 2018). Moreover, the authors conclude the paper with the following desiderata for future research:

“In future research, the reliability and validity of the Sefo and related interactive ToM tasks need to be investigated in systematic and in-depth ways, ideally in a multi-lab, preregistered collaborative endeavor such as ManyBabies (Frank et al., 2017).”

Given that we already know from previous studies that the results of the Sefo task have been mixed, and the ManyBabies project is already well underway, why continue to flood the market with failed replications that do not provide any conclusive evidence as to why the results of the original studies have not always replicated?

For what is worth, I want to copy here my review of another paper that this group published recently with Royal Society Open Science (led by Louise Kulke, who has made an entire career out of publishing failed replications – and failed replications alone), as I think it shows how the ‘publication bias’ that the authors seem so concerned about (see p. 38) is in fact a signature of their own work in recent years:

“Related to the last point, we should consider what would have happened if this study had been an original study, and not a conceptual replication of previous work. In view of their null results, the authors would have had to admit – as most of us have had to admit often enough – that their study had failed: the manipulations they had carefully introduced to tap a certain effect simply didn't work. However, while researchers addressing new questions with new paradigms face the risk of failing, researchers only aiming to replicate previous studies seem to have found a new business model in Academia: if after a few failed replications, researchers finally manage to replicate the original results, the replicated results would be news and therefore publishable material. However, if their manipulations failed (as they clearly did here), they can always write up a new failed replication and continue to question previous findings, while not adding anything new to the literature.

I think this practice is highly questionable as it makes failed replications immune to failure (ironically) and always publishable, regardless of the possible shortcomings that would prevent publication of original studies. Since publication standards should be just as high for original work and replication studies, I cannot recommend this manuscript for publication.”

Author's Response to Decision Letter for (RSOS-191998.R1)

See Appendix B.

Decision letter (RSOS-191998.R2)

Dear Mrs Wenzel,

It is a pleasure to accept your manuscript entitled "Actions do not speak louder than words in an interactive false belief task" in its current form for publication in Royal Society Open Science.

As was discussed, we will now approach the more critical of the reviewers to invite them to submit a Comment to highlight the contentious debate in the field. We will let you know if they choose to submit a Comment.

on behalf of Dr Teodora Gliga (Associate Editor) and Essi Viding (Subject Editor)
openscience@royalsociety.org

Associate Editor Comments to Author (Dr Teodora Gliga):

Thanks a lot for carefully addressing the reviewers and my comments and for the clarifications added to the manuscript. I believe it now offers a fair depiction of the existing literature and sufficient methodological detail to allow future studies to clarify variation in performance in these tasks.

Appendix A

Dear Dr. Gliga,

Thank you very much for your comments concerning our manuscript (RSOS-191998: "Actions do not speak louder than words in an interactive false belief task"). We are happy to hear that you found merit in our systematic approach and about the chance to revise the paper. We found your and the reviewers' comments and suggestions very helpful in improving the manuscript, and we here send you a carefully and thoroughly revised version of our manuscript that addresses these concerns. Please find our response to the comments below.

We were careful to provide a more balanced literature review and description of previous findings. In particular, we included an important review paper on the variety of successful studies on early ToM (Scott & Baillargeon, 2017) and two commentaries on the current replication crisis (Baillargeon et al., 2018; Poulin-Dubois et al., 2018), omitted the topic of replicability of priming tasks, and went more into detail in the description of some important studies (such as Prieuwater et al. (2018) or Carpenter et al. (2002)). Furthermore, we rephrased our descriptions of recent replication studies of the Sefo task in order to sound less critical and to provide more information on their rationale and implications. In addition, we paid special attention to methodological differences between our implementation and the original study. That is, we included detailed explanations and clarifications of our methods, and discussion on reasons for our replication failures. We also included additional analyses with alternative measures of early ToM in an appendix, and clarified in the discussion that children's performance in our standard TB task replicated a recurring pattern that is usually found in the literature.

In our view, the manuscript has substantially improved due to your and the reviewers' helpful comments. We thus hope you find the present version of our manuscript suitable for publication in Royal Society Open Science. We look forward to hearing back from you. Thank you very much.

Sincerely,

Lisa Wenzel
Sebastian Dörrenberg
Marina Proft
Ulf Liszkowski
Hannes Rakoczy

Reviewers' Comments to Author:

Reviewer: 1

Comments to the Author(s)

This manuscript reports three studies, attempting to replicate and extend an interactive paradigm by Southgate et al. (2010) that was meant to tap into infants' and toddlers' theory of mind abilities. Results using a close approximation of the original study failed to replicate its main finding (that 17-month-olds seemed to understand an actor's false belief), and the additional studies with various manipulations to attempt to make the task easier/more pragmatically felicitous also failed to demonstrate that toddlers (and even 3- and 4-year-olds) would respond in accord with an understanding of false belief. Adults, however, responded as expected on the "Sefo" task and their performance was correlated with other theory of mind measures.

This is a well-written paper, reporting a programmatic series of studies. I appreciated that the authors went beyond demonstrating a "failure to replicate" by attempting to modify the task still further and to investigate e.g., its relation to other tasks. I have a few comments and suggestions.

1. My primary comment is that I believe the authors have demonstrated conclusively that they were unable to replicate findings from the original task. But this leaves open the question (which I don't think the authors address, though I may have missed it) about why the discrepancy? Is it because of the small sample size in the original study? Were the authors able to confirm with the original researchers that the procedure was done in exactly the same way? As I know the authors are aware, failures to replicate can occur for many reasons and it would be good to be able to work with the original researchers to try to understand together why their findings were different.

Thank you very much. The reviewer is right that we did not sufficiently discuss reasons for the discrepant findings between our study and the original study. In response, we now discuss this at length in a new paragraph in the General Discussion (including the now following points, see p. 37-38)

First of all, we want to mention that we were in contact with Victoria Southgate (the first author of the original study; she was actually a research fellow and advisor in the research network in the context of which the present project was located), discussed the procedure of her Sefo task, and sent out videos of our procedure to her from our last replication approach (Dörrenberg et al., 2018) before we finalized the procedure for the present set of studies. She confirmed that we implemented "a pretty faithful replication" of her task. As a possible explanation for the discrepancy between our results, she pointed out that differences in the ostensive communication of the experimenter could reduce infants' engagement (personal communication). However, it is important to mention that in the current collaborative replication project, different persons served as experimenters within and between studies. Of course, each experimenter might have her own ways of interacting with infants – one might be better capable of communicating ostensively than the other – yet, none of them was successful in uncovering infants' precocious FB understanding and no differences between labs were found. In addition, during extensive, direct trainings with each experimenter, we made all efforts to ensure that the interaction-based procedure of the Sefo task was socially engaging and sound, and we were always careful in obtaining infants' cooperation, engagement and rapport during warm-up sessions in the reception room before testing, and maintaining it throughout the testing sessions.

We agree with the reviewer, that reasons for replication failures may be diverse. So, what else could be reasons for the discrepancy between our findings? It is possible that methodological alterations other than interaction style led to different results. Although we aimed in Study 1 at pursuing close direct replications, there may still be differences. For instance, we used a phrasing of the prompt that did not include any reference to the child's memory of previous events (to ensure that there is no linguistic hint to the non-referred container), and we did not use a novel word label (e.g., Sefo). In the original study, however, 17-month-olds succeeded with three different kinds of prompts that did or did not refer to memory, and did or did not include a novel label. In addition, in contrast to earlier studies (Carpenter et al., 2002; Happé & Loth, 2002), a recent study by Papafragou et al. (2017) found no performance enhancing effect of word learning over classical FB contexts, when both tasks types were equated. In line with these findings, in our Study 2a, where we used the phrasing of Southgate et al. from Experiment 3 (with a novel label), as well as in the study by Dörrenberg et al. (2018), where we used the phrasing of Southgate et al. from Experiment 1 (with reference to memory and a novel label), we still failed to replicate the original pattern. This raises the possibility that the phrasing of the prompt (e.g., the use of a novel label), may not be important for infants' success on the Sefo task. Another possible explanation for the discrepancy is that the original study overestimates the actual effect size, which is a common problem in psychological science. Only large-scale, collaborative replication projects may be able to demonstrate whether this is the case. In the course of the *ManyBabies* consortium, we are currently working together with Victoria Southgate (and many other original authors) in order to implement such a large-scale replication project of the Sefo and related interaction studies.

2. I would also encourage the authors to consider further the implications of their findings. They argue persuasively that whether toddlers understand false beliefs has important theoretical implications re nativism, for example. This seems reasonable. But it seems like overstating the case to suggest that theory of mind reduces to an understanding of false beliefs, which I don't think the authors want to argue but which is an interpretation a reader could make. Additionally, I'd encourage the authors to try to link the abilities they believe toddlers have or do not have to children's real-world experience. What would it mean (beyond performance on experimental tasks or theoretical debates about nativism) for children to have the kind of understanding that some scientists attribute to them?

Thank you very much for the suggestions. We agree that the human Theory of Mind has more facets than only false belief ascription that develop earlier in life (e.g., perception goal psychology, joint attention). However, subject of the presented discussion is the fully-fledged Theory of Mind, with meta-representations at its core.

Here, usually FB tasks serve as litmus tests: Individuals must understand that the agent's (false) belief about a given situation is incompatible with their own (true) perspective, and that both realities cannot coexist. Thus, if infants master these tasks, it would not only indicate that they understand FBs but possess a full-blown ToM with all related competences (e.g., understanding of aspectuality). Therefore, we put our focus on FB tasks and less on those aspects of ToM that develop earlier in childhood.

3. I wondered whether the authors could code from video other aspects of the children's performance that could also speak to some of the debates in the field. For example, were children slower to make responses in the TB than FB conditions (even though they ended up being at chance in which object they chose)? Did they look more at one than the other object/box depending on condition? What kind of spontaneous comments (if any) did children make—did they request clarification? What kinds of individual differences might obtain in these kinds of tasks?

Thank you for this suggestion. We agree that it is a very good idea to look for other potential indicators of children's tracking of the agent's belief in the Sefo task. In response, we now recoded the videos of the 17-month-olds for additional, potentially informative behaviors (although gaze behavior cannot be reliably coded from all our videos given the camera angles, and children did not request clarification) and report these additional results in the result section of Study 1 (p. 15) and the Appendix (p.44).

In addition, in Study 1, we coded whether infants warned E1 about the presence or the actions of E2 (e.g., by pointing at the curtains where E2 was hiding, or by indicating that objects have been swapped) which is a measure that has been used in previous interactive ToM studies (e.g., Knudsen & Liszkowski, 2012): of the 17-month-olds, only 4 of 24 (17%) in the FB condition and 3 of 24 (12.5%) in the TB condition (here E1 already knew about E2) did so in the first trial, which is statistically not different (Fisher's exact test, $p = .701$); of the 3-year-olds, 6 of 24 (25%) in the FB condition and none in the TB condition did so, which is significantly different (Fisher's exact test, $p = .022$). Of those six 3-year-olds that warned E1 in the FB condition, 5 correctly chose the non-referred container. There was a significant difference between those who warned or did not warn E1 for their choice of container (Fisher's exact test, $p = .007$). This effect cannot be found for the 17-month-olds and it is restricted to less than a quarter of 3-year-olds. However, it underscores that some 3-year-olds did track E1's knowledge state and responded appropriately (not simply by chance).

We now also coded latency of children to make an approach/a choice after the prompt for the 17-month-old's in Study 1. Similarly, we found no differences between TB and FB condition (Mann-Whitney $U = 207$, $Z = -1.671$, $p = .095$). Even though, infants took longer to decide for one box in the TB condition (8344 ms, $SD = 13682$) than in the FB condition (2858 ms, $SD = 2635$), this difference was statistically not significant. In addition, we coded the direction of infants' first look. We were able to extract this information from 39 videos reliably. In the remaining 9 videos, the camera angle was unfortunately not facing the child, or the video quality was too poor to identify the gaze direction. However, we again did not find any differences between the conditions (Fisher's exact test, $p = .556$). In both conditions, infants performed at chance level (binomial tests, $ps = 1$): 53% infants look to the referred box first in the TB condition (9 infants), in the FB condition 50% did so (11 infants).

Thus, unfortunately, none of these alternative measures provides conclusive evidence for early belief ascription. Concerning individual differences, there is currently not much information on whether predictors of later/explicit ToM, such as personality, language skills, executive control, parental interactions, SES, earlier socio-cognitive skills etc., also affect early ToM competencies.

Reviewer: 2

Comments to the Author(s)

Review of RSOS-191998

Entitled 'Actions do not speak louder than words in an interactive false belief task'

Overview:

This paper presents 3 experiments with the expressed objective to replicate (closely and with modifications) the study of Southgate, Chevallier and Csibra, 2010. The paper mainly focuses on the relevance of the replication approach and on the difficulties in finding robust and reliable methods for studying the development of theory of mind. No doubt, replicability is a central issue for research lead by the idea of accumulating valid knowledge, and psychology is an area of research facing a serious replication crisis. That said, it is extremely important to pursue transparent and stringent research practices in such replication attempts as well and at the same time the theoretical framing should remain objective.

I appreciate this general approach together with the efforts the authors invested into empirical testing, nonetheless, I have concerns with respect to their presentation of data and the conclusions that can be drawn from them. I miss important details in the presented methods and results and find the interpretation of the selected pattern of data somewhat biased.

In order to this paper become more stringent, persuasive and constructive, presentation of additional methodological details, data and a more toned down interpretation is proposed.

Concerns in details:

Introduction:

My main impression is that the authors handle replication problems and theory driven debates on the interpretation of given findings as the two facets of the same problem (the authors handle these as reliability and validity of the empirical phenomenon). While both questions have the highest importance, and clearly they are related, these issues cannot be directly equated. The entire introduction suffers from this latter issue (the validity of a phenomenon is not always handled correctly in my opinion). Bringing up replication attempts and their results as the main focus of the paper seems to justify the selection to include only those studies that has been targeted to be replicated. Clearly, this approach results in a selective set of references. This is fine unless someone does want to question the underlying theoretical constructs as well, in which case the related conceptual approaches (though not necessarily being replications) would be important to be mentioned and considered in details. Otherwise the reader receives a biased picture of the state of the art research in the field. So, if the authors' objectives entail the discussion of theoretical approaches as well, they ought to include the contrastive explanatory angle and the empirical bases of them too.

Let me highlight this problem with the help of some examples:

1. One of the the authors' main claims is:

'In fact, a growing body of replication studies puts into question the robustness of the original evidence on early and automatic ToM (for an overview, see Kulke & Rakoczy, 2018).' p. 5

Following this claim, the authors mention the failed replications of both anticipatory looking tasks and looking time studies, however they ignore the commentary that focuses on the problems with the above replication attempts, see Baillargeon, Buttelmann, Southgate, 2018. Also, the authors do not mention that a collection of conceptual replications are reviewed in the literature (see Scott and Baillargeon, 2017). The purpose of representing the debate itself - and not only their own take on - would necessitate discussing the other viewpoint, or at least referring to it.

We are grateful to the reviewer for drawing our attention to this issue. We did not intend to present imbalanced information or a biased view on the current situation in the field. It is important to note that providing an exhaustive literature review on early ToM or diving deep into the global debate on replicability of original evidence goes beyond the scope of our current paper, since we focused exclusively on the replicability and validity of a specific task. Therefore, throughout our paper, we were careful to present our findings as failures to replicate the task (absence of evidence) and not as a lack of competence (evidence of absence) in our participants. However, after repeated inspection of the introduction, we agree that we missed out on mentioning some important papers. We now cite the review paper by Scott and Baillargeon (2017) on the variety of different measures and successful conceptual replication studies on early ToM (see p. 4), and we further mention the ongoing debate on the severity of the replication crisis, represented by Baillargeon et al.'s commentary on methodological issues with current replication studies, that the reviewer suggested, but, to stay objective, also the response commentary by Poulin-Dubois et al. (2018), that takes a different stance (see p. 4).

2. Here is another example on what kind of phrases and claims I find misleading: 'Similarly, ToM-based priming effects in implicit tasks with adults could recently also not be replicated (Conway, Lee, Ojaghi, Catmur & Bird, 2017).' p. 5.

This sentence suggests that implicit ToM-based priming tasks in general are not replicable. There are several problems with this claim then. First, a closer look at the referred paper reveals that the objective of the study is to offer a competing interpretation for an empirically reliable (replicable) phenomenon. So, while providing a conceptual replication of the phenomenon, the paper aims to disentangle different explanations beyond it. That is true, the findings seem to support a non-mentalistic interpretation of the data, yet this study is NOT an exemplar of a non-replication. Second, even if the role of the study in the paper was to represent problems with replicating similar studies, the authors should not dismiss that the actual phenomenon has been replicated by a couple of other studies conceptually, with contrasting interpretation to the one mentioned (see Furlanetto et al., 2016; Marshall et al. 2018; Wiese et al., 2012; Teufel et al., 2010), and there are other paradigms on perspective taking with similar theoretical background that are actually highly replicable across laboratories (Surtees et al., 2016; Elekes et al, 2016.)

We thank the reviewer for the comments concerning our description of replications of priming tasks. Since the focus of the current study was the replication of an infant ToM task, where interpretations regarding the scope of the underlying ToM capacities are still in dispute (they could be automatic/implicit, or even mature and based on conscious processes), and since most priming tasks focus on automatic ToM capacities in adults, which is a topic for itself, we decided to omit this issue from our manuscript (see p. 4).

3. With respect to interactive FB tasks, the manuscript focuses on the so called helping task by Buttelmann et al., 2009 and the attempts to replicate it. The argument in the manuscript is that while this empirical paradigm is at least partially replicable, there is a problem with the validity of the original task, as it 'may not measure belief-tracking but simpler forms of social cognition such as goal-understanding.' p. 6

I am fine with the claim that the responses given in the experimental protocol can be interpreted differently, however the dispute would be mentioned in more details, to make obvious for the readers in what sense the validity of the protocol is in question. I am emphasizing this as the alternative interpretation itself also received criticism, namely that the teleological approach mentioned can explain how children bring up reasons for an already performed behavior (not necessarily 'a simpler form of social cognition'), but based on that approach there is no answer on how and whether children are able to predict the behavior of a protagonist -the original problem of ToM research (see <http://cognitionandculture.net/blogs/pierre-jacob/if-teleology-is-the-answer-what-was-the-question/>).

We appreciate the reviewer's approach to discussing scopes and limits of alternative interpretations. However, apart from any theoretical notion, but thinking in the most objective and parsimonious ways, passing the Buttelmann task does not require participants to engage in false belief representation, which is the key competence in this debate. Accordingly, Prieuwater et al. (2018) showed in their study that infants' helping behavior in the Buttelmann task was not guided by an evaluation of the agent's belief about the object location, but rather by the agent's goal to reach the object. That is, Prieuwater et al. added another control condition to the Buttelmann task, in which the agent in the false belief condition unsuccessfully tried to open a third box that was always empty (neutralizing the agent's false belief). As in the original false belief condition, in this new condition, infants still helped the agent by opening the box containing the object. We now rephrased this paragraph, explain the study by Prieuwater et al. in more detail, and refer to Baillargeon et al.'s commentary (2018, p. 115-118) for concerns about replication failures and alternative interpretations of the Buttelmann task (see p. 5).

4. Arriving to the focal purpose of the paper, the authors introduce the sefo task. I miss here the detailed description of the Loth and Happé task (2002) and also of the task of Carpenter et al (2002). Though done with 3 year olds, those studies served as sources for the sefo task and actually provide convergent evidence that 3 year-olds are able to exploit belief ascription for the sake of word learning.

Thank you. We now give more information on the studies by Happé & Loth and Carpenter et al. (see p. 5-6). However, we now also mention that a recent study by Papafragou et al. (2017) found no performance enhancing effect of word learning over classical FB contexts, when both task types were equated, raising the possibility that such effects in the literature derive from mismatched control conditions. It is important to note

that even the study by Southgate et al. (2010) found no difference in performance with or without word learning.

5. In relation to the sefo task the already existing replication attempts are discussed. Here, I would like to suggest a careful and thorough description and interpretation of data in general, as this is the most helpful strategy to support open science practices and the understanding of the robustness of the phenomenon.

So, with respect to the Grosse Wiesmann et al. (2017) study, it would be important to mention that the main question in such a task is whether the response pattern in the FB condition shifts away from children's natural (baseline) response to a request accompanied by a pointing gesture which is measured in the TB condition. Therefore, a comparison between TB and FB performance is needed in order to grasp the modulatory effect of belief ascription in the interaction, that is actually missing in the above experiment (while that is true that the original study could also show above-chance performance for the FB conditions).

Thank you. We already mention that Grosse Wiesmann et al.'s study only tested FB conditions and that the replication failed in terms of a comparison to the original above chance performance (page 7, lines 164-166). We now explicitly state as a limitation of that study, that a corresponding TB condition is missing (see p. 8). Yet, since 3-year-olds in the study by Grosse Wiesmann et al. performed below chance in the FB condition of the Sefo task, a direct comparison to a TB condition might not even be conclusive.

With respect to the Kiraly et al (2018) study the authors raise: 'a main concern is that this finding is difficult to interpret given that no analogous prospective TB condition was conducted.' I do not see the main concern here. Actually, there was a TB condition that was used as a bases for comparison for the prospective FB condition as well: at the beginning of the protocol infants explored a pair of sunglasses that were see through, and then they saw the model using an other pair of sunglasses. True, that in this condition children got themselves information that those pair of sunglasses the model wore were transparent as well. Is there any theory that would predict difference in performance just based on receiving confirmatory information later? In what sense a 'just prospective TB condition would modify the results? So, this concern is of no base, it is simply misleading.

Thank you for raising this point. Indeed, the argument we made is redundant in that Király et al. present data of the retrospective TB condition. Since no belief revision takes place in this scenario, there is no need of an analogue perspective TB condition to compare the performance to the FB condition. Nevertheless, we would like to state that matched control conditions, that are as closely as possible to the test conditions, are best scientific practice (especially in times of replication crises). However, we now omit this sentence from the paper (p.7).

Clearly true that this conceptual replication contained modifications that were difficult for young children in the specific TB-FB condition in terms of memory load, where 18-mo-olds failed, but that was the target question of the paper of Kiraly et al. (children had to keep track of the status of the sunglasses and relate that to the possible belief content of the protagonist with the help of their memory capacities). What is the relevance then of the description of this modification in the present paper without mentioning the target question of the original paper? Again, this is somewhat misleading, as it is presented as a concern.

We are afraid that this is based on a misunderstanding. It seems we did not make our points clear enough. In response, we clarify and explain much more carefully and explicitly in the revised version: We do not criticize that study because of the modifications, since these were intended and necessary to test the target question. The modifications, however, make it hard to interpret Király et al.'s findings (the positive and the negative ones) in light of replicability of the original task (which by the way they claim to have done in their paper), since we do not know which factors might have boosted or diminished performance. We now made this clearer and also highlighted Király et al.'s research question (see p. 7).

In addition, both for the Grosse Wiesmann study and the Kiraly et al. study the argument is brought up that the absence of the deceptive motive could have led to the low performance in the FB condition. Is it the claim that just based on the fact that the model remained in the room during the test in both studies the deceptive mode is questionable? In my opinion, as actually the deceptive mode was performed in both studies while the model

was present, the authors should also try to give an explanation on how children interpret the deceptive mode in the presence of the model. So, the suggestion is either give more details on this explanation or leave out the post hoc interpretation, as it cannot give a systematic explanation (same context resulted in failure in 3- and 4-year-olds in Grosse Wiesmann et al study, failure in 18-mo-olds in Kiraly et al study, yet in success in 3-year-olds in Kiraly et al study).

Thank you for raising this point. We now left this argument out for the Kiraly et al. study, since this is a conceptual replication with lots of differences to the original study (p. 7). Grosse Wiesmann et al. discuss in their paper that the missing deceptive motive in their task could have decreased performance – and we therefore mentioned it, too. Yes, these are post-hoc attempts at explaining diverging results – but all of the attempts to explain the massively diverging findings from the original studies, the one successful replication and the several non-successful ones are, by necessity, post-hoc. They way out, of course, is the pre-planned collaborative replication project envisaged under the umbrella of *ManyBabies*.

The concerns with respect to the linguistic complexity used for older children are also somewhat confusing. If I understand correctly, the argument is that in case of the study of Kiraly et al., the use of the original phrases from Experiment 1 of Southgate et al. (Do you remember what I put here?) would have induced a pragmatic shortcut for 3-year-olds to solve the task without belief ascription. But only for the FB condition, since, according to the authors: ‘due to different pragmatic contexts between conditions, children may interpret the prompt in the TB condition differently’. What are those differences in the pragmatic context (for the TB and FB condition) that are completely unrelated from belief ascription? Please describe the details, I do not find them obvious. Rather, this claim would predict pragmatic ambiguity for the TB condition as well if children do not take into account the belief of the speaker. (Note again, the original study did show the same pattern of results with three different prompting).

We argue that the prompt used in Kiraly et al. is not a pragmatic shortcut, but potentially simply a semantic one. With the phrase “Do you remember what I put in here? [...]” E1 refers to the object that is now in the other box. Since the objects were swapped, the correct answer to this question would always be to choose the object out of the other (non-referred) box independent of E1’s state of belief. While this semantic explanation leads to the pattern found in the FB condition, it would not predict the pattern found in the TB condition. Here, maybe other plausible strategies may come into play, e.g. to obey the pointing gesture without paying attention to E1’s request. Since she witnessed everything that has happened, the situation is very clear for E1 and the child. In the FB condition, in contrast, children may also get confused, when trying to keep track of the events E1 misses during of her absence. Note that these alternative interpretations do not include belief ascription at all. We now make differences in the TB condition clearer in the paper (p. 8-9).

Study 1: Direct replication

Methodological issues

1. I find compelling that ‘40% of the 17-months-olds tested in Study 1 struggled to pass the original familiarization criterion’. p 16. The authors therefore decided to a) modify the procedure slightly for some participants, and b) applied a more lenient passing criteria for a subsample of participants. One can find a sample description in the result section, referring to the lenient criteria passers: FB: n = 12, TB: n = 8. So then, the sample of children who passed the original criterion (FB: n = 12, TB: n = 16) includes those participants for whom the procedure was modified slightly or not? This particular detail could be of importance to evaluate the discussion of the difficulties of familiarization in relation to the reliability of the phenomenon. The authors argue for the reliability of the protocol by emphasizing that there was no performance difference between the subsamples that did vs. did not reach the familiarization criterion. This claim would receive more emphasis if the subsample of passers would not include those who received modified demonstrations.

Thank you very much for raising this point. We now report how many infants received the modified familiarization version and that both subsamples (original and lenient passers) can include these infants (p. 14, table 1). In addition, we would like to point out that the exact procedure of the familiarization phase is not well described in the original paper. Important information is missing (where boxes open/closed, did E1 only ask for objects or did she point at the boxes) that even Victoria Southgate could not faithfully provide us in personal communication. However, she told us that she “think[s] if you opened the boxes first, or pointed to the boxes, it should serve the same function” as closed boxes and only asking for the object. Our modified version of the warm-up phase included open lids during the request and thus, resembles the warm-up phase described in Grosse Wiesmann et al. (2017).

Table 1: Number of children that chose the referred/non-referred box in the first test trial with the applied warm-up criterion (original or modified) as a function of the warm-up version (original or modified)

		Original criterion	Modified criterion
TB	Original warm-up	2/1	1/0
	Modified warm-up	8/5	5/2
FB	Original warm-up	2/2	1/0
	Modified warm-up	5/3	7/4

(The claim that for 3-year-olds the exact same familiarization did not provide any challenge while they also failed in the test is inconclusive already because of the potential age differences: the fact that there was no problem for older children does not rule out that there could have been issues for younger ones. Nonetheless, actually there was a tendency difference in the performance in TB vs FB conditions in the older age-group).

2. There are some missing minor methodological details by the clarification of which the methods would be more clear. When did E2 introduced the deceptive mode? Already at the beginning of the location swap episode? Was this deception mode there for the TB condition as well? How long was E1 away? How many pointing were shown in the test phase (just one or more)?

Thank you for pointing out the missing details when describing the deceptive mode. We provide now more details in the method section of study 1 in line with the original study (p. 11).

3. While the authors explicitly declare that Study 1 is a direct and as close as possible replication of the original study they modify the verbal prompts used in the task. The verbal prompts of Experiment 2 and Experiment 3 of the original study were combined. The result is: “Do you know what’s in here? I want to play with this”... “Can you give it to me?”. (Just for comparison, verbal prompts of the original study: Exp 2 of Southgate et al.: Do you remember what I put in here? Shall we play with it? Shall we play with it? Let’s play with it! Finally, she said, ‘Can you get it for me?’ Exp 3: Do you know what’s in here? There’s a sefo in the box! There’s a sefo in here! Can you get it for me?)

The authors themselves call attention to the potential effect of the pragmatic ambiguity of some verbal prompts that can overshadow children’s belief ascription capabilities, (likewise they argue for the case of the prompt used by Grosse Wiessman et al, 2017; see page 8.) I myself think that while the purpose of this new prompt could have been to use the most neutral phrasing, the result of the choice is again equivocality. The assumption beyond the interactive FB task is that word label mapping or reference mapping trigger belief ascription: in search for a correct mapping of the referent of the interlocutor’s request the child need to track the belief state of her. However, what is the message of the prompts used in this study? The model asks whether the CHILD knows what is in the box, whatever it is, and clearly, she expresses she would like to play with that object. In contrast, the versions of the original study highlight the possible mismatch between the protagonist’s perspective and the child’s knowledge, either by referring to a previous event and a previous state of the scenario (using the phrase ‘Do you remember what I put in here?’) or by introducing a novel label, for the mapping of witch the child needs to track the protagonist mental states (she cannot know that the object in here is the SEFO). So, the pragmatic call of the phrase in the present version is different from that of the originals.

I think it is highly relevant to acknowledge this difference in the paper. Just saying that this prompt is similar to the original version of Exp 3 is inaccurate. Based on this, it is questionable to call the study as direct replication.

Indeed, we neither referred to a previous scenario nor used a novel label in the prompt in Study 1 (but see Studies 2a and 3). Importantly, as already mentioned, a reference to previous events in the prompt provides a semantic shortcut to the non-referred box without belief ascription - it should therefore be avoided. That is why we decided to use the phrase “Do you know what is in here?” from Southgate et al.’s Exp. 3, where it produced the same results as in Exp. 1 (“Do you remember what I put in here?”) even though it asks “whether the CHILD knows what is in the box, whatever it is”, as the reviewer pointed out. If the reviewer would be right, it should have led infants to choose the referred box also in the original study – but it did not. Concerning word label mapping, the study by Southgate et al. showed that infants did not perform less proficiently without (Exp. 2) than with a new word label (Exp. 1 and 3). In line with this, a recent study by Papafragou et al. (2017) calls into question even the basic effect of an advantage of word learning contexts on toddler’s performance in

referential FB tasks (as originally found by Carpenter et al. (2002) and Happé & Loth (2002)). Accordingly, it is by no means a fact that word label mapping triggers FB tracking, and it is especially unclear whether it affects infants' processing in implicit tasks (since there is no controlled study showing this). Nevertheless, taking a systematic look on the prompts used in original and replication studies, it seems that including one part or the other (or both) does not ensure successful FB tracking in the Sefo task. In Study 2a, for example, we used the novel label without a reference and were still not able to find the original pattern. While Grosse Wiesmann et al. (2017) used a prompt with reference to previous events and a novel label ("Do you still know" + Sefo), Dörrenberg et al. (2018) even used the original phrasing of Exp. 1 ("Do you remember" + Sefo). Yet, both studies failed to replicate the original findings of children choosing the non-referred box. Actually, our Study 2b aimed at ensuring belief tracking from the pragmatic context, detached from the verbal prompt, yet still unsuccessfully. We now clarify that our prompt in Study 1 is a combination of Southgate et al.'s prompts from Exp. 2 and 3 (p. 12), describe the original prompts in more detail in the Introduction (p. 6), and we discuss this prompt issue in the General Discussion (p. 37-38).

4. The study introduced three consecutive trials for testing 17-mo-olds where feedback was given. The text describes the forms of feedback for the FB conditions, but what was the feedback for the TB conditions?

Thank you for pointing out the missing information. The feedback was adjusted for the TB condition: If the child chose the object out of the referred box, E1 acted happily; if the child chose the object out of the non-referred box, she acted in surprised and sad ways. We now report the feedback in general with additional information on the TB condition (p. 12)

5. It is unclear from the text that in the case of the 3-year-old group each child was tested in only TB conditions or FB conditions? That is what the following sentence means? „3 -year-olds were tested with a single Sefo FB/TB test trial and one trial of a corresponding standard FB/TB (change-of-location) task (task order counterbalanced).“

Thank you. Children were either tested in a Sefo FB task and a corresponding standard FB task, or in a Sefo TB task and a corresponding standard TB task. We agree that the description of the study design was not clear and describe it now more detailed (p. 11).

Results

1. What is the exact reason for collapsing the sample and check whether there is a tendency to choose the referred box?

Thank you. The reasons for this calculation were post-hoc analyses to check whether there is a general tendency to choose the referred box over all conditions. Since we did not provide any hypotheses and/or predictions about these analyses, we omit these now (p. 13).

2. Can the authors report a matrix of failers/passers with respect to the sefo and the standard task? If I understand correctly, this data is available, though only separately for TB and FB conditions.

Thank you for this suggestion. We now provide a contingency table with children's performance in both tasks on page 16 (table 3).

			SBT	
			empty box	box containing the object
Sefo	TB	non-referred box	1	1
		referred box	12	10
	FB	non-referred box	3	4
		referred box	9	8

3. I doubt that a correlation analysis is well grounded when the data for the to-be-correlated measures on both sides show random pattern!! At least, no correlation in such case is definitely inconclusive.

This seems to be based on a misunderstanding. The fact that children as a group perform at chance level in two tasks is perfectly compatible with substantial correlations between the tasks (if one sub-group of children tends to pass while another sub-group tends to fail both tasks consistently). Therefore, correlation analyses are an appropriate means to see whether a systematic picture emerges amongst different developmental milestones.

Discussion

1. Is there any explanation on why there was no learning effect in the TB condition for 17-mo-olds?

Since we did not find learning effects in the FB condition, we would not expect one in the TB condition. And majority (67%) was correct in first trial TB, thus, they did not get feedback.

2. What does the standard TB version measure?

Once established as a structurally analogous control task to the standard FB tasks, the standard TB task was supposed to measure children's belief ascription abilities in a situation in which no conflicting beliefs exist (in contrast to FB tasks where the child's and the protagonist's belief are divergent). However, recent studies show a more complex picture of children's development of TB understanding (Oktay-Gür & Rakoczy, 2017; Fabricius, Boyer, Weimer, Carroll, 2010; Perner, Huemer & Leahy, 2015). Once young children start to master FB tasks, they begin to fail TB tasks with massive negative correlations. In several follow-up studies, Rakoczy & Oktay-Gür (2020) showed that pragmatic confusion seems to mask children's true TB performance. We now elaborate more on this in the discussion (p. 17).

3. In discussion of the performance of the 3-year-olds, (who actually performed at chance in both standard FB and TB tasks), instead of accepting that children were simply guessing in both kinds of tasks, the interpretation is raised that there are two separate subsamples (FB passers/TB non-passers and TB passers/FB non-passers in the group of 3-year-olds. This interpretation sounds very speculative. Is there any solution to prove it based on the data? Otherwise, based on Occam's razor, (and also based on the data on TB non-passers introduced in the paper they cite) it is more likely that the study represents a no replication of the TB version of the classic task.

Actually, the interpretation of the TB performance of the 3-year-olds is not speculative at all, but based on the converging evidence from a large set of recent studies (Oktay-Gür & Rakoczy, 2017; Rakoczy et al., 2020, Perner et al., 2015). As mentioned above, young children pass TB tasks and fail FB tasks, until the pattern reverses around age 4 with failure in TB tasks and success in FB tasks. Only from age 8-10 do children then consistently master both FB and TB tasks. Thus, children's performance in TB tasks shows a U-shaped curve between age 3 to 10 years (until they master both kinds of tasks) with strong negative correlations to their performance in FB tasks. Further findings of Oktay-Gür & Rakoczy (2017) and Rakoczy & Oktay-Gür (2020) suggest that children from age 4 are pragmatically confused by the trivial test question in TB scenarios: Here, the experimenter asks the child to predict the behavior of an agent, who is informed about every relevant detail of the situation. So, if everyone witnessed what has happened (e.g. that the object's location has changed), why is she asking? Pragmatically modified TB tasks (e.g. by highlighting the pragmatic context of the test question) eliminated children's confusion and revealed a TB performance as proficient as their FB performance.

We added more information about the interpretation of TB results to the discussion section of Study 1 (p. 17) and also to the general discussion (p. 36).

Study 2a

Methodological issues

1. Note, the pragmatic ambiguities revealed by the verbal prompts hold fold the 'original' sefo task here as well, as described above.

In study 2a we use the exact prompt of Exp. 3 of the original study (“Do you know what is in there? There’s a Sefo in that box! There is a Sefo in here! Shall we play with the sefo? Can you get it for me?”) in both task versions original and modified. We agree with the reviewer that this is not described sufficiently, and we now report the method of study 2a more precisely (p. 20-21).

2. Actually, in the modified task the verbal prompting is different, it contains a new verbal labelling for one of the new objects as in Exp3 of Southgate et al. However, based on the description, the experimenter continuously pointed at the box during the verbal prompting. This is again a difference from the original study. Was there a difference in this respect between Study 1 and the ‘original’ trials of Study 2a? Worth to mention this detail, as this strong referential cue can overwrite children’s knowledge based inferences (as the authors themselves discuss this possibility in the general discussion, but not as part of giving the rationale for Study 2b).

Thank you for pointing out further possible interferences and differences between the setups. From the original paper, it is not clear when and for how long exactly the pointing took place. However, in videos that we received from Victoria Southgate the pointing lasted as long as the experimenter was verbally referring to one of the boxes (“Do you remember what I put in *here*? There’s a Sefo in *here*. There is a Sefo in *this* box.”) and we adjusted our procedure accordingly in all the studies presented in this paper. Thus, the pointing in our studies cannot be considered as deviant from the original study. To be more precise, our pointing lasted as long as the verbal reference to one of the boxes, independent of the request’s type. We now describe the requests and pointing more carefully in every method section (e.g. p. 20).

3. Were there any feedback or related verbal description between the trials?

No, we did not give any feedback after the first trial, since we did not want to influence children’s performance in the second trial. Thank you for pointing out missing information. We now report more details on Study 2a on (p. 21).

Results

Clearly, in all conditions children followed the pointing behaviour of the experimenter and chose the object from the referred box. I think that there is no reason to calculate correlation in case of such homogenous pattern at ceiling.

We agree that this analysis does not provide any information and omit the analysis now from the paper (p. 22).

Study 2b

I find the objective and the implementation of this task very interesting. Ecologically it seems to be more valid, the request of the experimenter emerges from an obvious need. However, I still have doubts whether this implementation is pragmatically more transparent than the previous ones. In my reading, the familiarization of the task assures children that the experimenter knows what makes an apparatus work. When the experimenter is back and deliberately referring to one of two boxes -each of which contains a novel object -, and says ‘We need what is in there. Can you give it to me?’, this can be taken as an assertion that they (the child and the experimenter) need the specific object that is in the box the experimenter actually is referring to. As previously she turned out to be knowledgeable, no doubt emerges in the situation.

I would like to underline again that in previous versions a tension between the referential cuing and the assertion was present (‘remember’ or the need to map a novel label to an object).

Thank you very much for this issue. This was exactly what we were aiming for: ensuring that the pragmatic context makes clear that the Sefo is a specific object (that is needed for the game) and that therefore it is important where E1 believes that it is. And the question “We need what is in there. Can you give it to me?” has exactly the referential ambiguity required here: On the required pragmatic reading of that questions, what the agent is referring to is the object she takes to be in the box (rather than the object that is in the box).

Study 3

The question remains open in what context infants literally follow the strong referential cuing and when they are sensitive to the belief of the model. Study 3 proposes to investigate this question. The general introductory argument is that there was a repetitive failure in finding evidence that the sefo task is measuring whether young children keep track of other agents' beliefs in communicative interactions. The theoretical perspective can be formed differently: introducing modifications to the sefo task may alter the interpretation of the ongoing interaction, and consequently, the interpretation of the communicative intention of the partner by infants: in most of the cases they are led by the teaching intent of the protagonist, when it occurs within highly ostensive cues. The discussion and introduction would benefit from raising this alternative (Csibra and Gergely, 2009).

Thank you for your comments on Study 3. We would like to clarify again the main aim of the third study, which was not to investigate the circumstances in which infants choose to follow ostensive cues over their counterpart's belief, but rather whether the Sefo task really measures belief ascription. Since the interactive Sefo task is also a modified version of explicit referential belief tasks, the question remains open, whether children interpret the Sefo task in same ways than the explicit referential belief tasks. We found in Study 1 and 2a that the original age group (17-month-olds) and 2-year-olds did not track the protagonist's perspective, even though we implemented the original setup (with changes in the prompt, but not in interaction). Comparisons to matching explicit task versions tested with a control group possessing ToM competences could provide evidence for the intended measured ability and therefore the task's validity. Therefore, we implemented the explicit test questions of a referential belief task into the procedure of the Sefo task to directly compare children's performance in the different task versions interactive and explicit.

There is no clear description on what the explicit question versions of the Sefo task intended to measure. Indeed, they offer a possibility to answer the question whether children reacted differently in the behavioural request situation and in the label and belief questions. As those answers are coming from the same participants, a related sample analysis is suggested: that could show that there were differences in the answer patterns for the different tasks, thus potentially different interpretation of the request and the questions on the part of the same participants. I think the McNemar test fails to grasp the change between choices within participants.

Thanks for pointing out that we missed to mention the purpose of the explicit test questions.

As mentioned above, the interactive Sefo task was adapted after different explicit referential belief tasks. As a result of combining several procedures and implementing them into an interactive setup, it is not clear whether the intended abilities are measured. Therefore, we implemented both questions label and belief according to the explicit referential belief task by Papafragou et al. (2017) in the setup of Study 3a. We now refer to our aims and the debate when introducing the explicit questions (p. 29).

Regarding the McNemar's tests, there seems to be a misunderstanding: This test is exactly the method of choice for comparing within-subjects response patterns across two tasks with dichotomous measures. More specifically: In order to determine children's changes in their response between the referred to the non-referred box (and vice versa) from one explicit task to the other, the McNemar test should be the appropriate statistical test, and the same applies to the comparisons to the explicit FB task.

Still, there is no discussion on the fact that the same children intended to follow directly the referential request of the experimenter, while their belief mapping and labelling attempts were significantly different from their behavioural choice. What is the explanation for this?

Thank you for pointing this out. In the General Discussion, we suggest that the pointing gesture could explain children's weak performance in the Sefo task, and produce the pattern that children performed even better after explicit test questions. This suggestion arises from several studies that found a robust bias to search in pointed-to locations (e.g., Palmquist & Jaswal, 2012). We now add this possible explanation for the different performances between the interactive and the explicit Sefo tasks to the discussion section of Study 3a (p. 35-36).

General discussion

I completely agree with the authors that the Sefo task in the future need to be more systematically investigated and tested in multilab collaboration. However, before arriving to this conclusion, it would worth to summarize

that even in the present paper there were some inconsistencies in result patterns (chance level performance for 17-month-olds for both FB and TB sefo conditions in Study 1, tendency difference for 3-year-olds between FB and TB conditions in Study 1; always choosing the referred box irrespective of condition at ceiling in Study 2; difference in the pattern of responses for behavioural request and the verbal questions in the sefo trials in Study 3) that highlights that the sefo task as being an interactive task, the focus of interaction can be shifted sensitively to contextual factors.

Thank you for your suggestions. All in all, our findings are not as inconsistent as presented when taking a look at the original interactive Sefo task. While 17-month-olds chose at chance level (Study 1), 2-year-olds chose the referred box at above chance levels indiscriminately in both FB and TB condition (Study 2), as did the 3;6-4;6-year-olds (Study 3). For the infants in Study 1, the explanation might be that they in fact chose randomly due to extraneous performance factors and procedural ambiguity. The only inconsistency is the age group of 3-year-olds that chose at chance level in the FB condition in Study 1, and, together with 4;0- to 4;6-year-olds, predominantly the referred box in Study 3. This is also the only age group where we found a trend towards a significant difference between the conditions. We did not find in any of the other age groups differences between conditions. However, to further check whether there are developmental consistencies within the age group of 3-year-olds we merged the data of 36- to 48-months-old children, that were tested with a FB condition either in Study 1 (24 children) or Study 3a (11 children). Within this subsample of 35 three-year-olds, 27 children chose the referred box while 8 decided for the non-referred box. This performance differed significantly from chance level (binomial test, $p = .05$) and shows 3-year-olds preference for the referred box. These results are thus much in line with the other age groups. We added this new analysis to the result section of Study 3 (p. 32) and the summary of findings in the different age groups to the General Discussion (p. 37).

The problems with the standard TB task should also be mentioned in the general discussion.

Thank you for this suggestion. We now include a section on the TB findings in our general discussion (p. 36).

The discussion of the factor that the pattern of data in Study 2 and Study 3a is similar to the tendency found in the literature that 3- to 4-year-olds exhibit a robust bias to search in pointed-to locations even when the pointer has turned out to be unreliable, deceptive or ignorant about the object location deepens the need to study further what influence the behavior of children in such interactive tasks.

References:

- Baillargeon, R., Buttelmann, D., Southgate, V. (2018). Invited Commentary: Interpreting failed replications of early false-belief findings: Methodological and theoretical considerations. *Cognitive Development* 46. 112-124
- Csibra, G., Gergely, Gy. (2009) *Natural Pedagogy*. *Trends in Cognitive Sciences* 13(4):148-53
- Elekes, F., Varga, M., & Király, I. (2016). Evidence for spontaneous level-2 perspective taking in adults. *Consciousness and Cognition*, 41, 93–103.
- Furlanetto T, Becchio C, Samson D, Apperly I. (2016). Altercentric interference in level 1 visual perspective taking reflects the ascription of mental states, not submentalizing. *J Exp Psychol Hum Percept Perform.* 42(2):158-63.
- Marshall, J., Gollwitzer, A., Santos, L.R. (2018). Does altercentric interference rely on mentalizing?: Results from two level-1 perspective-taking tasks. *PLoS One*, 13(2), e0194101. doi: 10.1371/journal.pone.0194101
- Scott, R. M., & Baillargeon, R. (2017). Early false-belief understanding. *Trends in Cognitive Sciences*, 21(4), 237-249.
- Surtees, A., Apperly, I., & Samson, D. (2016). I've got your number: Spontaneous perspective-taking in an interactive task. *Cognition*, 150, 43–52.
- Teufel, C., Alexis, D. M., Clayton, N. S., & Davis, G. (2010). Mental-state attribution drives rapid, reflexive gaze following. *Attention, Perception & Psychophysics*, 72(3), 695–705.

Wiese, E., Wykowska, A., Zwickel, J., & Müller, H. J. (2012). I see what you mean: How attentional selection is shaped by ascribing intentions to others. *PloS One*, 7(9), e45391.

Appendix B

Dear Dr Gliga,

Thank you again for your comments on our manuscript and the opportunity of an additional revision. We are sending you here the revised version of our manuscript and our responses to the reviewers.

We have now made it clearer throughout the paper that our studies refer exclusively to FB understanding and not to early socio-cognitive skills involved in fully-fledged ToM development. Additionally, we have revised our abstract, so that we now present the full picture of the replication situation (mixed results) and corrected the introduction regarding theoretical accounts. We now also provide a table presenting the original study and all studies that directly and conceptually replicated the Sefo task including age group, sample size, and results.

However, one thing we would like to clarify, in response to a point made in your letter (requesting revision), is the following: our findings of older children (3- to 4-year-olds) performing at chance level in the standard FB task are *not anomalous*, but rather very much in line with traditional evidence. In Wellman et al.'s (2001) meta-analysis, for example, children around the age of 48 months passed explicit FB tasks with a proportion below 60%. In addition, Wimmer and Perner (1983) found in their original FB task that children between 4 and 5 years predominantly failed to correctly solve the explicit FB task. Since our sample included 3- and 4-year-old children (with a mean age of 47;9 months in Study 3a), they performed as expected at chance level.

We hope the revised manuscript is now suitable for publication and thank you again for the helpful comments. We look forward to hearing back from you.

Sincerely,

Lisa Wenzel
Sebastian Dörrenberg,
Marina Proft,
Ulf Liszkowski,
Hannes Rakoczy

Reviewer: 1

Comments to the Author(s)

This revision reports an attempted replication and extension of research on false belief understanding in toddlers. I think the revised version has addressed several issues and I appreciate that the authors have explicitly offered suggestions for why their findings differ from those of the original study. I offer a few additional comments below.

1. I continue to have concerns that the authors are conflating “theory of mind” with an understanding of false belief. For example, on p. 4 of the revision, they write, “a growing body of replication studies puts into question the robustness of the original evidence on early ToM.” But as noted in my review of the original manuscript, it seems to me that what the authors have concerns about is specifically the “robustness of the original evidence on early FB”. I recognize that the authors believe that FB is a “litmus test” for ToM. But I’m not sure how widely held this perspective is: It seems to me that most folks seem to believe that ToM involves a suite of social cognitive skills; an understanding of FB is but one (a point the authors seem to make on p. 5 when they attribute findings from Buttelmann et al., 2009, to “goal-understanding”). One solution would be to make clear it what the authors are objecting to and arguing by e.g., changing “ToM” to “FB” throughout (which would also match the title, which is specific to false belief).

Thank you for your comment. Indeed, with terms like “early ToM” we refer to an early understanding of false belief. To be clearer on that, we took the reviewer’s advice and changed “ToM” to “FB understanding” whenever it made sense.

2. The explanation of the Sefo task was unclear in the distinction between Study 1 and Study 2. I think for Exp. 2, it would be helpful to provide the entire language used to make it clear what was the same and what was different from Study 1. That is, I’m inferring that in Study 2, the E said, “Do you remember what I put in here? Shall we play with it? Can you get it for me?” But the “Do you remember what I put in here” is not included in the text for Study 2.

Thank you. For study 2a we state on p. 22 that we used the request of experiment 3 of the original study, which does not include “Do you remember what I put in here?” but “Do you know what is in here?”. The request of study 2b is adjusted to the pragmatically modified procedure and does not resemble one of the original requests.

3. The order of events in the procedure described for Study 2 on p. 21 could be clearer. That is, the information about what happens after the swap in terms of how E1 asks for objects is provided before the information about what happens before the swap in terms of how E1 puts the objects in the boxes and E2 changes their positions.

Thank you for noting. We changed that (p. 22).

Reviewer: 3

Comments to the Author(s)

'Actions do not speak louder than words in an interactive false belief task'

This manuscript reports three experiments trying to replicate (Study 1), extend (Study 2) and validate (Study 3) the results originally observed by Southgate and colleagues using the so-called 'Sefo task' (2010). While I did not review the first submission, I could see from the authors' response letter that R1 and R2 had raised important points and offered detailed and careful comments. In my review, I will address a major concern I have about Study 1, point out a couple of smaller points, and close with a general comment about failed replications.

[1]

In Study 1, which was supposed to be a direct replication of the original study, the authors had problems getting infants to pass the warm-up trials, to the point of having to relax the inclusion criteria used by Southgate et al., and in some cases leave open the boxes for the infants to reach for the objects (see p. 10). When discussing the results of their first study, the authors acknowledge this shortcoming (see pp. 17-18). However, they rule out the possibility that this issue could explain their null results on the following grounds:

"We did not find performance differences between the subsamples that did vs. did not reach the familiarization criterion. Furthermore, all three-year-old children passed the original familiarization criterion and still performed at chance level in the FB condition. Therefore, it seems unlikely that the low performance in the familiarization trials led to the low performance in the test scenario and thus the failed replication of the original results."

While the above arguments are sound, this discussion fails to acknowledge a more worrying possibility: infants' poor performance in the warm-up trials could reveal a lack of engagement in the task, or even a lack of rapport with the experimenter (something that would be key to pass this task, since participants need to understand the experimenter's goal). If this problem was general enough, infants could have underperformed across the board, which would explain their chance performance (note that the infants in Study 1 did not show a preference for the referenced box, unlike the other age groups). The above statistical comparisons, which the authors use to defend the validity of their findings, do not address such a problem.

I suggest that the authors acknowledge that infants' poor performance in the warm-up trials (something that was not observed by Southgate et al. (2010), nor in Studies 2 and 3 in this paper) could be symptomatic of problems in engaging the infants in the task, which could explain their chance performance without casting doubts on the validity of Southgate et al.'s results.

We have added a discussion of this, p. 18. We do not agree with reviewer 3 that infants' poor performance in the warm-up trials could reveal a general and symptomatic lack of engagement in the task that would explain their chance performance. We adjusted the warm-up phase and the passing criterion, because e.g., some of the infants needed a lot of time to approach one of the boxes. To decrease memory demands, boxes thus were left open during and after the request.

However, it is important to note that all infants interacted with the experimenter in the test trials and, thus, were engaged in the task. That is, infants responded to the experimenter's prompting by approaching a box – the majority of infants even followed her pointing irrespective of warm-up performance (and test condition). However, even if we would attribute a lack of engagement to the subsample of infants with poor warm-up performance, there is no reason to assume that this also applies to the subsample with good warm-up performance. The latter group brought the requested object in two consecutive warm-up trials and interacted with the experimenter in three of three test trials, clearly indicating that they were engaged in the task, at least as engaged as in the original study. Crucially, at least in part, the same applies to the “poor warm-up performance” subsample, since they also brought the requested object in two warm-up trials (just not in consecutive ones) and interacted with the experimenter in three of three test trials. So, all in all, we cannot see any convincing reason to assume that infants were generally less engaged in the present study.

[2]

In addition, Study 1 is reported as a ‘direct replication’ of Southgate et al. (2010). However, as the authors acknowledge on p.11, they mixed different features of the original experiments, which in my view doesn't qualify as a direct replication. I appreciate that Study 2 departed more radically from the original design, but I still think it's confusing to refer to Study 1 as a direct replication when it did not use the protocol of any of the original experiments.

As we acknowledged and discussed in our paper, the prompt of the experimenter differed slightly between our Study 1 and the original studies from Southgate et al. (2010). Importantly, the rest of our protocol was identical to the original protocol. As explained in our paper, we implemented this “new” prompt (actually a combination of the prompts used in Southgate et al.'s experiments 2 and 3) mainly to ensure that older children's choice of box was not biased by low-level hints, but would rest on their ability to attribute false beliefs. Crucially, our decision to implement a deviating prompt rested on the findings by Southgate et al., who showed in the course of three experiments that infants' choice of box in the Sefo task was not depending on what the experimenter was saying (certainly because they do not even fully understand it at 17 months). That is, in the original study, 17-month-olds succeeded with three different kinds of prompts that did or did not refer to memory, and did or did not include a novel label. On the other hand, there is evidence that the original prompt phrasings do not even guarantee success on the Sefo task: Dörrenberg et al. (2018) failed to replicate the Sefo task with the original prompt from Experiment 1; in Studies 2a and 3 of the current paper, we failed with the original prompt from Southgate et al.'s experiment 3.

In general, whether a replication is direct or conceptual is certainly a matter of degree. What we want to emphasize is that Study 1 is (close to) a direct replication, whereas the other studies, in the spirit of conceptual replications, introduced some (motivated) modifications to see whether children's performance could thus be boosted. To signal this more clearly, we could, if you wish, call Study 1 something like “(Close to) direct replication”.

[3]

More generally, I think the authors should acknowledge that they have had problems getting their participants to meet the original inclusion criteria of the studies they have tried to replicate. The low

performance observed in warm-up/familiarization trials could be indicative of an overall lower reliability of the critical trials – a possibility that is not refuted by comparing the performance of children who pass and fail the inclusion criteria.

Baillargeon, Buttelmann and Southgate already addressed this issue in their 2018 commentary in *Developmental Psychology*:

"Table 1 reveals clear differences between the rate at which participants (infants, older children, and adults) met this inclusion criterion between the different versions of the paradigm. For example, whereas 64% of 25-month-olds exhibiting an anticipatory saccade in Southgate et al. (2007) correctly predicted the location of the agent's search by the second familiarization trial, only 25% and 34% of infants did this in Schuwerk et al. (2018) and Grosse Wiesmann et al. (2018), personal communication), respectively. Similarly, Senju et al. (2009) and Low and Watts (2013) both reported that 100% of their adult participants (and for Low & Watts, also 3- and 4-year-olds) showed correct prediction by the last familiarization trial, whereas only 45% of adult participants reached this criterion in the Kulke et al. (2018b) study. These low inclusion rates meant that in order to reproduce the original analyses, these papers had to make inferences based on samples that were of a comparably small size to the original study (and in one case from 9 infants, Kulke et al., 2018a).

To what extent these differences have implications for the interpretation of test trial failures is unknown. When all (Low & Watts, 2013; Senju et al., 2009) or most (Senju et al., 2010; Senju et al., 2011; Southgate et al., 2007) participants pass the second familiarization trial, one could conclude that as a group, they understood the task and were motivated to make a prediction. When only half the sample exhibit a correct anticipatory look on the second familiarization trial, it is legitimate to ask whether what looks like success in these 50% is actually success, or rather just random looking. The lights in the window will draw participants' attention to one of the windows so there is a 50% chance of being 'correct' without anticipating action. Thus, it is not straightforward to use equivalent test trial failure in those who passed the second familiarization trial and those who did not as justification for abandoning the original inclusion criteria (Dörrenberg et al., 2018). To compare, in their meta-analysis of the traditional false-belief task, Wellman et al. (2001) excluded from their analysis any studies in which fewer than 60% of children answered the comprehension- check control questions correctly."

The quotation from Baillargeon et al.'s commentary that reviewer 3 provided actually refers to a different task, the anticipatory looking paradigm from Southgate et al. (2007). This revision is not a place for discussing that task's familiarisation and inclusion rates, since the whole purpose and procedure is not comparable to the Sefo task. We want to refer to the response commentary by Poulin-Dubois et al. (2018) for an elaborate discussion on replications of the anticipatory looking task. Only a general note: If test trial performance does not differ between participants who meet vs. do not meet an arbitrary inclusion criterion, there might be something wrong with the inclusion criterion in the first place. However, in case of the Sefo task, the warm-up with infants simply serves the purpose "to familiarize them with searching for objects in the boxes" (Southgate et al. 2010, p. 908) – it should therefore not be overstated. Note that in contrast to the Sefo task, other comparable interactive false belief tasks with infants do not even administer warm-up trials or execute inclusion criteria based on warm-up trials (e.g., Buttelmann et al. 2009). Please see point [1] for discussion on infants' warm-up performance in our Study 1.

[4]

I understand from the reviewers' comments that the original manuscript came across as biased. While I think the authors have done a good job in revising the manuscript, the abstract continues to be one-sided: the authors only refer to failed replications of the Sefo task, failing to acknowledge that the results with that task have been mixed.

Thank you very much for calling our attention to missing information in the abstract. We now changed that (p. 2).

[5]

The literature review includes an inaccuracy on p. 4:

“These findings on infants' FB understanding and adults' automatic tracking of others' perspectives had far-reaching implications and laid the grounds for novel theoretical approaches to ToM development, including nativism (Scott & Baillargeon, 2017) and two-system-accounts (Apperly & Butterfill, 2009).”

Nativist accounts of Theory of Mind have been around since the birth of the field in the 80's. In fact, Onishi and Baillargeon (2005) motivated the first successful study with infants as an investigation of those early nativist accounts (citing Alan Leslie and Jerry Fodor, amongst others).

While I appreciate that the authors want to emphasize the implications of false-belief studies with infants for the field, they should not overstate them.

Thank you for noting. We changed the phrasing (p. 4).

[6]

Finally, I would like to raise the question of what exactly this study contributes to the literature. As the authors now acknowledge in the introduction, three recent studies have already shown that the results of the Sefo task have been mixed (Grosse Wiesmann et al., 2017; Dörrenberg et al., 2018; Kiraly et al., 2018). Moreover, the authors conclude the paper with the following desiderata for future research:

“In future research, the reliability and validity of the Sefo and related interactive ToM tasks need to be investigated in systematic and in-depth ways, ideally in a multi-lab, preregistered collaborative endeavor such as ManyBabies (Frank et al., 2017).”

Given that we already know from previous studies that the results of the Sefo task have been mixed, and the ManyBabies project is already well underway, why continue to flood the market with failed replications that do not provide any conclusive evidence as to why the results of the original studies have not always replicated?

For what is worth, I want to copy here my review of another paper that this group published recently with Royal Society Open Science (led by Louise Kulke, who has made an entire career out of publishing failed replications – and failed replications alone), as I think it shows how the 'publication

bias' that the authors seem so concerned about (see p. 38) is in fact a signature of their own work in recent years:

“Related to the last point, we should consider what would have happened if this study had been an original study, and not a conceptual replication of previous work. In view of their null results, the authors would have had to admit – as most of us have had to admit often enough – that their study had failed: the manipulations they had carefully introduced to tap a certain effect simply didn't work. However, while researchers addressing new questions with new paradigms face the risk of failing, researchers only aiming to replicate previous studies seem to have found a new business model in Academia: if after a few failed replications, researchers finally manage to replicate the original results, the replicated results would be news and therefore publishable material. However, if their manipulations failed (as they clearly did here), they can always write up a new failed replication and continue to question previous findings, while not adding anything new to the literature.

I think this practice is highly questionable as it makes failed replications immune to failure (ironically) and always publishable, regardless of the possible shortcomings that would prevent publication of original studies. Since publication standards should be just as high for original work and replication studies, I cannot recommend this manuscript for publication.”

We do not provide an answer to this (in our view, highly inappropriate) comment at this point, but refer to our e-mail that we sent to the editors after receiving these lines.